# MVMP-HMR: MULTIVIEW MULTI-PERSON HUMAN MESH RECOVERY UNDER LARGE SCENES WITH OCCLUSIONS

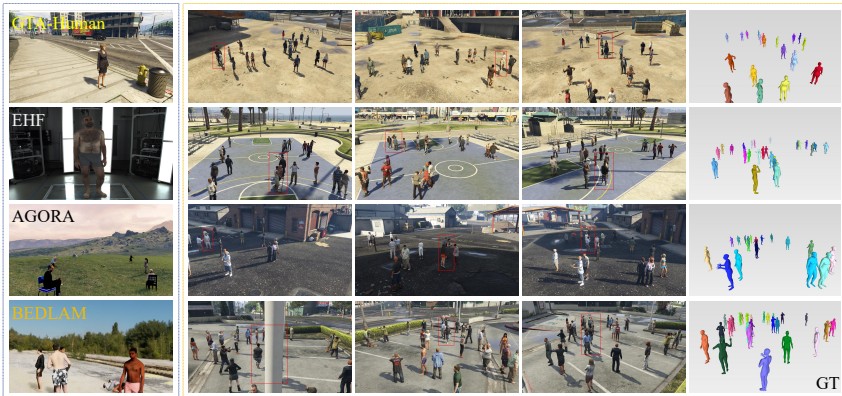

Figure 1: Comparison between single-view HMR datasets and our proposed MVMP-HMR dataset. The left shows images from GTA-Human (Cai et al., 2024b), EHF (Pavlakos et al., 2019), AGORA (Patel et al., 2021), and BEDLAM (Black et al., 2023) datasets from top to bottom. Our multiview images and ground truth meshes are shown on the right, containing larger scenes and more persons. The red box indicates areas with severe occlusions.

## ABSTRACT

Human mesh recovery (HMR) refers to recovering the human 3D meshes from images. Most existing HMR tasks focus on multi-person from a single image or a single person from multiple views. And the evaluation benchmarks used in these methods usually contain quite small numbers of humans or under small scenes, which is unreliable for real applications with severe occlusions. Thus, we present Multiview Multi-Person HMR (MVMP-HMR), a multiview model for multi-person whole-body human mesh recovery from multi-view images under occluded scenes. Specifically, MVMP-HMR first fuses multiple views to obtain a 3D feature volume for all persons, and then the pelvis joint from a 3D pose estimation net is utilized to acquire the human query of each person from the 3D feature volume. Finally, the human queries are cross-attentioned with the 3D feature volume and integrated to decode each person's 3D meshes. Besides, two novel losses are put forward to further enhance the model performance: the orientation loss and the 3D joint density loss, dealing with the orientation and pose ambiguities in the mesh predictions under the occluded scenes. Furthermore, a large synthetic MVMP-HMR dataset is proposed, which consists of 15 multiview scenes with up to 50 camera views and 30 persons. Experiments demonstrate that the existing state-of-the-art (SOTA) HMR methods cannot perform well on the proposed large MVMP-HMR benchmark, and the proposed MVMP-HMR model's advantages over existing SOTAs under large scenes with severe occlusions.

# 1 INTRODUCTION

Human mesh recovery (HMR) predicts the human 3D meshes from images or image crops, which has important applications in autonomous driving, digital games, or AR/VR, *etc*. Most existing HMR methods focus on recovering human meshes for scenes with a quite limited people number (usually $< 15$ in total), either with a single person from single images or multi-crops, or multi-persons from single images. Besides, the evaluation benchmarks used in the latest methods are usually under small scenes, with few occlusions (see Figure 1 left). This is not practical for real-world applications where there might be massive crowds in large scenes with severe occlusions. Thus, the existing HMR methods have not been evaluated under more complicated conditions with both larger human numbers and more severe occlusions, whose performance is not ensured.

To solve the problem and extend the HMR task to more complicated scenes, in this paper, we present MVMP-HMR (as in Figure 2), a novel model for multi-person whole-body human mesh recovery from multi-view images, which fuses multiview clues to handle the severe occlusions in large scenes with more humans. Specifically, MVMP-HMR extracts single-view features and projects them to the 3D space, and then the projected multi-view features are averaged to obtain a complete 3D feature volume for the whole scene. Besides, a 3D pose estimation branch is adopted to predict the pelvis joint location of each person, and the predicted pelvis joint is used to acquire the human queries by sampling at the locations from the previously fused 3D feature volume. Then the human queries and the 3D feature volume are both fed into the human transformer block (HTB) where both are fused via cross-attention layers. Finally, the output of HTB is decoded to regress the SMPL-X parameters.

To deal with the human orientation and pose ambiguities in the predicted SMPL-X parameters under the occluded scenes, in addition to common parameter regression losses used in single-view HMR SOTA (Baradel et al., 2024), we put forward two novel losses: the **orientation loss** and the **3D joint density loss**. The orientation loss $\mathcal{L}_{\mathcal{O}}$ is the supervision of the human mesh's orientation in the real-world coordinates. The 3D joint density loss $\mathcal{L}_{denj3d}$ supervises the 3D joints in the predicted human mesh via 3D joint density maps instead of direct joint coordinate regression. Both provide stronger supervision in the 3D space and handle the orientation and pose ambiguities in the MVMP-HMR task better, further enhancing the model performance (see results in Sec. 4.5). Furthermore, we also propose a large synthetic multiview multi-person HMR dataset that contains more people, more camera views, and scene variations (see Table 1 for reference) compared to existing datasets.

In summary, the contributions of the paper are:

- As far as we know, this is the first study on the multiview multi-person HMR task under large scenes with severe occlusions. No existing research has focused on the issue in the HMR area. Besides, we propose a large MVMP-HMR dataset for studying the topic.
- We propose the MVMP-HMR model, which is the first multiview multi-person HMR model for reconstructing multiple persons with multiple views under large scenes. In addition, we propose two novel losses for better MVMP-HMR performance.
- Experiments demonstrate that existing methods cannot perform well under the new multiview multi-person HMR benchmark with severe occlusions, and the proposed MVMP-HMR method outperforms both existing single-view HMR state-of-the-arts (SOTAs) and 3D HPE with multi-view settings.

# 2 RELATED WORK

**Single-person HMR.** Human mesh recovery (HMR) predicts the human 3D meshes from images. The early HMR methods were based on optimization, and they were easily stuck at local minima (Hasler et al., 2010; Lin et al., 2023; Moon et al., 2022; Pavlakos et al., 2019). Instead of estimating the human meshes as in 3D reconstruction, (Kanazawa et al., 2018) proposed to predict SMPL parameters of the shape and 3D joint angles to represent human meshes from a cropped image. SMPLify-X (Pavlakos et al., 2019) followed SMPLify to estimate the 2D joints and optimize model parameters to fit them, and then improved over SMPLify with a new DNN trained on a larger dataset. In addition, many regression-based methods were proposed (Cai et al., 2024a; Choutas et al., 2020; Feng et al., 2021; Moon et al., 2022; Rong et al., 2021; Zhang et al., 2023; Zhou et al., 2021), which is focused on single-person estimation. Furthermore, many methods tried to utilize multi-crops to

Table 1: The statistics of the proposed MVMP-HMR dataset, Single-view HMR, and 3D HPE datasets. MVMP-HMR dataset contains more persons, more scenes with multiviews, and more complexities.

| Task | Dataset | Area | SceneNum | Subjects | Occlusion | Views | Frames | GT Format |
|---|---|---|---|---|---|---|---|---|
| Single-view HMR | GTA-Human | - | - | 1 | Simple | 1 | 1.4M | **SMPL**, J3D |
| | EHF | - | - | 1 | Simple | 1 | 100 | **SMPLX**, J3D |
| | AGORA | - | - | 5∼15 | Medium | 1 | 17K | **SMPLX**, SMPL, Mask |
| | BEDLAM | - | - | 1∼10 | Medium | 1 | 380K | **SMPLX** |
| 3D HPE | Human3.6M | 4mx3m | 7 | 1 | Simple | 4 | 3.6M | **SMPL**, J3D, Depth |
| | 3DPW | - | - | 1∼2 | Simple | 1 | 51K | **SMPL** |
| | CMU Panoptic | 5.49mx4.15m | 1 | 3∼8 | Medium | 65 | 1.5M | J3D, Depth |
| MVMP-HMR | Ours | 30mx30m | 15 | 10∼30 | Severe | 50 | 63K | **SMPLX**, J3D, Mask, Depth |

enhance the HMR performance (Choutas et al., 2020; Feng et al., 2021; Moon et al., 2022; Lin et al., 2023; Cai et al., 2023). In addition, HeatFormer (Matsubara & Nishino, 2025) is a neural optimization method based on 2d heatmap generated from SMPL parameters. *In summary, single-person HMR is limited to images with few persons, making it impractical for real-world scenarios with multiple people, larger scenes, and severe occlusion.*

**Multi-person HMR.** Compared to single-person HMR, multi-person HMR (Choi et al., 2022a; Goel et al., 2023; Kolotouros et al., 2019; Qiu et al., 2022; Zhang et al., 2021a) needs to predict the human meshes of multiple persons in the images. Multi-person HMR usually adopts a two-stage procedure: detect all humans in the image first (He et al., 2017; Liu et al., 2016; Redmon et al., 2016), and then perform HMR (Kim et al., 2023; Ma et al., 2023; Yoshiyasu, 2023; Zheng et al., 2023) for each detected person with crops. The two-stage process is not end-to-end and the occlusion in images may hurt the human detection accuracy, thus limiting the whole pipeline's performance. In contrast, single-stage methods have also been proposed (Sun et al., 2021; Qiu et al., 2023; Sun et al., 2022). Recent methods Multi-HMR (Baradel et al., 2024) and AiOS (Sun et al., 2024) adopted the DETR architecture for multi-person human mesh recovery. Multi-HMR (Baradel et al., 2024) detects 2D people locations using features of a ViT backbone and predicts their whole-body pose, shape, and 3D location using a cross-attention module. AiOS (Sun et al., 2024) performs human localization and SMPL-X estimation in a progressive manner, which consists of body localization, body refinement, and a whole-body refinement stage to regress SMPL-X parameters. Beyond mesh recovery, multi-view human analysis includes pose estimation methods (Zhang et al., 2020; Dong et al., 2021; Zhang et al., 2022) that focus on sparse keypoints, and avatar-based approaches (Lu et al., 2024; Lee et al., 2025) utilizing specific priors for high-fidelity reconstruction. Even though existing multi-person HMR methods can accurately estimate human meshes for several persons in single images, they are only evaluated on small scenes containing a small number of persons, eg, < 15. It is not clear whether they can be applied to scenes with larger sizes and severe occlusions. Thus, we propose MVMP-HMR, which fuses multiple camera views to deal with severe occlusions. *As far as we know, this is the first study for multi-person HMR with multiviews, and we also propose a large synthetic MVMP-HMR dataset, which shall advance the HMR task to more complicated conditions.*

**Single-view HMR and 3D HPE Datasets.** While numerous datasets have been proposed for Human Mesh Recovery (HMR) and other 3D human tasks (*eg.*, 3D Human Pose Estimation (HPE)), they have distinct human number, area size, and environmental complexity limitations compared with our dataset, as shown in Table 1. *Single-view HMR Datasets* like GTA-Human (Cai et al., 2024b), AGORA (Patel et al., 2021), and BEDLAM (Black et al., 2023) all employ synthetic data generation through game engines, and EHF (Pavlakos et al., 2019) is collected in the laboratory. Though providing SMPL-family parametric labels, they fundamentally suffer from depth ambiguity in monocular capture and lack real-world scene complexity. The number of people appearing in their scene is quite small, mostly just one person or at most 15 people in the scene, which is not practical in the real outdoors. Besides, since their scenes are quite simple with no other obstacles in the environment, the occlusion levels of the scenes are quite low. Therefore, existing HMR datasets are mainly based on single-view images, which are not applicable to more complicated scenes with large sizes and severe occlusions. *Compared to Single-view HMR datasets, our MVMP-HMR dataset provides a greater variety of views and a larger number of people. So, MVMP-HMR is more applicable in severe occlusion scenes.*

*3D HPE Datasets* include Human3.6M (Ionescu et al., 2013), 3DPW (Von Marcard et al., 2018), and CMU Panoptic (Joo et al., 2015a). While they capture real-world data through camera arrays

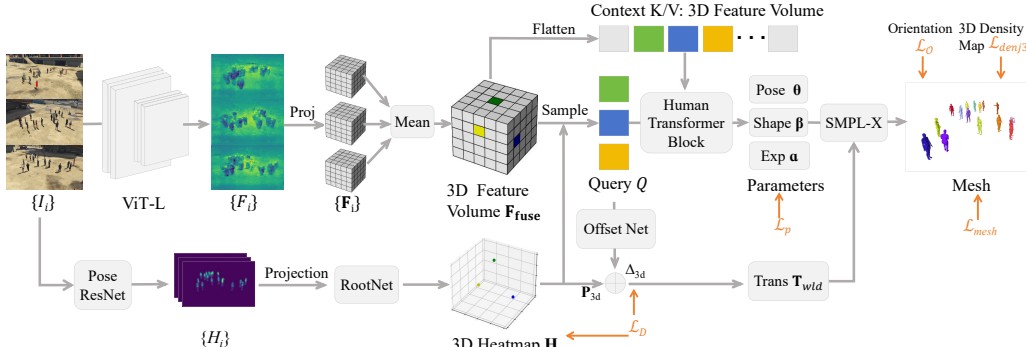

Figure 2: The pipeline of our proposed MVMP-HMR method, which consists of 3 main steps: Single-view Feature Extraction, Multi-view Feature Projection and Fusion, and 3D Decoding. We first extract single-view features with a ViT backbone, and then the single-view features are projected to the 3D space and averaged to obtain the 3D feature volume of the whole scene. Finally, with the joints outputted from a 3D pose estimation branch (at the bottom), we select pelvis joint to extract human queries for each person and feed them into a human transformer block (HTB) for 3D decoding and SMPL-X parameters prediction. In addition to losses previously used in single-view HMR SOTAs, we also put forward two novel losses, $e.g.$, orientation loss $\mathcal{L}_{\mathcal{O}}$ and 3D joint density loss $\mathcal{L}_{denj3d}$, for better orientation and pose accuracy in meshes.

and mocap systems, they still have three key limitations: 1) Limited human count: Typically each scene contains $\leq 10$ subjects, failing to represent crowded real-world environments; 2) Constrained scene sizes and type: They are all captured in studio environments or small indoor spaces ($leq50$ $m^2$), lacking large-scale outdoor variations, their background type is limited in the indoor scene, and they cannot cover the light or time change outdoors; 3) Simplistic occlusion patterns: Due to the limited number of people, they primarily contain light inter-person occlusion. *3D HPE datasets have a fixed environment setting, while our MVMP-HMR dataset can simulate changes in lighting and provide more expansive scenes. MVMP-HMR also offers more extensive annotation than 3D HPE Datasets. These strengths make our dataset more representative of real-world scenarios and better suited for practical applications.*

## 3 MULTIVIEW MULTI-PERSON HMR (MVMP-HMR)

We now introduce our multiview multi-person whole-body human mesh recovery task. Given multiview input RGB images $\mathbf{I} = \{I_1, I_2, \ldots, I_C\}$ ($C$ is the view number), our model (denoted as $\mathbf{f}$), directly predicts a group of $N$ centered whole body SMPL-X parameters such as pose $\theta \in \mathbb{R}^{N \times 53 \times 3}$, shape $\beta \in \mathbb{R}^{N \times 1 \times 10}$, and expression $\alpha \in \mathbb{R}^{N \times 1 \times 10}$, along with their associated 3D spatial translation $\mathbf{T}_{wld} \in \mathbb{R}^{N \times 1 \times 3}$ in the world coordinate system. It outputs expressive 3D human meshes $\mathbf{M} = \mathbf{SMPL\text{-}X}(\theta, \beta, \alpha, \mathbf{T}_{wld}) \in \mathbb{R}^{N \times 10475 \times 3}$:

Compared to single-view human mesh recovery (Single-view HMR), MVMP-HMR task obtains human meshes with absolute locations in 3D world coordinates, rather than relative positions in the camera-view coordinates, because single-view prediction has depth, orientation, pose, and occlusion ambiguities. Thus, MVMP-HMR utilizes multiple views for better multi-view fusion and multi-person mesh recovery to deal with these ambiguities and severe occlusions in practical applications. We require the multiple cameras to be calibrated and synchronized in the setting. As in Figure 2, the proposed MVMP-HMR model consists of three modules: Single-view Feature Extraction, Multi-view Feature Projection and Fusion, and 3D Decoding, whose details are as below.

### 3.1 SINGLE-VIEW FEATURE EXTRACTION

Our MVMP-HMR framework employs the Vision Transformer-Large (ViT-L) (Dosovitskiy et al., 2021) architecture as the backbone single-view feature extractor: $F_i = \mathbf{ViT\text{-}L}(I_i)_{i \in \{1,\ldots,C\}}$, where $i$ denotes the view id, $F_i$ denotes the feature map of view $I_i$, and $C$ is the number of views. To validate backbone selection, we conduct comprehensive experiments comparing various transformer-based architectures, with detailed ablation studies presented in the Appendix A. The ViT-L model

demonstrates superior performance in capturing global contextual features critical for multi-view fusion. Thus, we use ViT-L as the feature extractor.

In parallel with the ViT-L backbone, we use an HRNet (Sun et al., 2019) for 2D pose heatmap predictions $H_i$. After the single-view feature extraction, we obtain feature maps $\{F_i\}$ and heatmaps $\{H_i\}$ of all views. They are forwarded to the next step for fusion.

## 3.2 MULTI-VIEW FEATURE PROJECTION AND FUSION

The extracted single-view features are projected to a constructed 3D volume for multiview feature fusion. The constructed 3D volume size is $300 \times 300 \times 20$, each voxel dimension representing 100mm in the physical 3D world. So the volume's spatial dimensions are $30m \times 30m \times 2m$ in the real world. In the feature projection, we employ perspective geometries to map each 3D voxel coordinate $\mathbf{p}_w = (x, y, z)$ to 2D image coordinates of multiple views: $\mathbf{p}_c^{(i)} = \mathbf{K}^{(i)}[\mathbf{R}^{(i)}|\mathbf{t}^{(i)}]\mathbf{p}_w$, where intrinsic $\mathbf{K}$ and extrinsic $[\mathbf{R} \mid \mathbf{t}]$ matrices are provided in the MVMP-HMR dataset, and $i$ denotes the camera view index. We project each view's feature map $F_i$ into a 3D volume through this perspective-aware coordinate projection, and each view's 3D feature volume is denoted as $\mathbf{F}_i$. Then, we fuse the projected multi-view feature volumes via a mean operation, and the fusion result is denoted as $\mathbf{F}_{\mathbf{fuse}}$.

2D heatmaps $H_i$ are projected into a 3D volume, then fed into a modified RootNet (Tu et al., 2020) to generate 3D probability heatmaps $\mathbf{H}$ (encoding pelvis joint likelihoods in world coordinates). Fusion of these heatmaps yields the coarse 3D grid location $\mathbf{P}_{\mathbf{3d}}$ of the primary (pelvis) joint.

## 3.3 3D DECODING

The fused 3D feature volume $\mathbf{F}_{\mathbf{fuse}}$ is decoded with a Human Transformer Block (HTB) to regress the SMPL-X parameters in the 3D world. For each detected human $n \in \{1, ..., N\}$ in the 3D heatmap $\mathbf{H}$, we use pelvis joints to sample human features $q$ from $\mathbf{F}_{\mathbf{fuse}}$. Then we combine $q$ with $X$ to construct human queries (denoted as $Q$), and $X$ denotes the mean SMPL-X model parameters. Besides, the 3D feature volume $\mathbf{F}_{\mathbf{fuse}}$ is flattened as one-dimensional vectors as Keys and Values. Then we input Queries, Keys, and Values into our HTB for SMPL-X parameter regression.

Figure 3 shows the details of the Human Transformer Block. The full flattened vectors are used as cross-attention keys $K$ and values $V$. The hu-

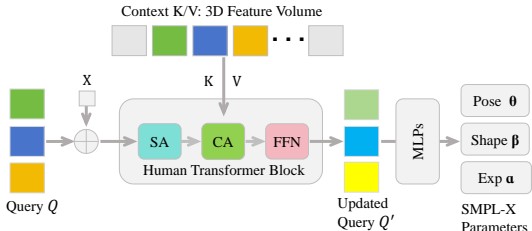

Figure 3: The details of the HTB: human queries are updated first via the self-attention layer (SA), the cross-attention layer (CA) integrated with flattened 3D features, and the FeedForward (FFN) layer, and then decoded via MLPs for SMPL-X parameter regression.

man queries $Q$ are updated with a stack of $D$ HTB. Then, three MLPs are introduced to regress each human's SMPL-X parameters $\theta$, $\beta$, and $\alpha$ with the updated human queries $Q'$.

Human queries $Q$ are also fed into a 3D offset prediction net to estimate the offset $\Delta_{3d}$ of humans. Combining the primary joint location $\mathbf{P}_{\mathbf{3d}}$ in the 3D heapmap and $\Delta_{3d}$, we can get the final location of the human's primary location, denoted as translation $\mathbf{T}_{wld} = \mathbf{P}_{\mathbf{3d}} + \Delta_{3d}$. Finally, we input the SMPL-X parameters and the translation $\mathbf{T}_{wld}$ to the SMPL-X layer (Pavlakos et al., 2019) for acquiring humans' mesh vertices and joints locations in world and camera view coordinates.

## 3.4 TRAINING LOSS

Overall, we adopt five losses to train the proposed MVMP-HMR model. The first three types of losses are similar as in the prior work (Baradel et al., 2024): the **detection loss** for localizing the human queries, the **SMPL-X parameter regression loss**, and the **mesh loss** for supervising 3D joints and vertices coordinate regression in human mesh format. Besides, since our task is in the 3D coordinates system, with orientation and pose ambiguities under the occluded scenes, we propose two novel losses to further enhance the model performance: the **orientation loss** for better orientation prediction

instead of the direct SMPL-X parameters predictions, and the **3D joint density loss** supervising the predicted 3D joints from the human meshes in 3D density format instead of direct 3D joint coordinate regressing. The details of each loss are as follows.

**Detection loss**. With the help of the heatmap prediction branch **HRNet** (Sun et al., 2019), we can get the 3D heatmap $\mathbf{H}$ of the primary joint of each human in the scene. Then we construct a 3D volume to present the occupancy of people as $\hat{\mathbf{H}}$ with GT joints location. We also obtained the 3D offset $\Delta_{3d}$ in the grid to get a more refined coordinate. So we have the detection loss $\mathcal{L}_D$ as follows: $\mathcal{L}_D = ||\mathbf{H} - \hat{\mathbf{H}}||_2 + |\Delta_{3d} - \hat{\Delta}_{3d}|$. where $\hat{\mathbf{H}}$ and $\hat{\Delta}_{3d}$ are the ground truth 3D heatmap and location offset of the joints, respectively.

**Parameter regression loss**. All SMPL-X parameters predicted by the model are computed with $L_1$ regression losses. We integrate the body model parameters (pose $\theta$, shape $\beta$, expression $\alpha$) into loss function as follows: $\mathcal{L}_p = |\theta - \hat{\theta}| + |\beta - \hat{\beta}| + |\alpha - \hat{\alpha}|$, where $\hat{\theta}, \hat{\beta}$, and $\hat{\alpha}$ are the GT parameters.

**Mesh loss**. After predicting SMPL-X parameters, we can construct human meshes from a SMPL-X layer. Then we extract 3D joints $J_{3D}$ and vertices $V_{3D}$ from the human meshes and project these 3D points onto the 2D multi-image planes. The mesh loss supervises the 3D/2D vertices and joints:

$$\mathcal{L}_{3D} = |J_{3D} - \hat{J}_{3D}| + |V_{3D} - \hat{V}_{3D}|, \mathcal{L}_{2D} = |\pi_i(J_{3D}) - \pi_i(\hat{J}_{3D})| + |(\pi_i(V_{3D}) - \pi_i(\hat{V}_{3D})|, \quad (1)$$

where $\hat{J}_{3D}$ and $\hat{V}_{3D}$ are the ground truth 3D joints and vertices, $\pi_i$ is the camera projection operator, and $\pi_i(\hat{J}_{3D})$ and $\pi_i(\hat{V}_{3D})$ refer to the ground truth 2D joints and vertices projected from the 3D ground truth. And the mesh loss $\mathcal{L}_{mesh}$ combines the two losses: $\mathcal{L}_{mesh} = \lambda_1 \mathcal{L}_{3D} + \frac{1}{C} \sum_{i=1}^{C} \mathcal{L}_{2D}$. Loss weight $\lambda_1$ adjusts the weight for the two loss terms and we use a fixed value $\lambda_1 = 100$ in all experiments. *In addition to these losses, we propose two novel losses:*

**Orientation loss**. The global orientation (a low-dimensional vector) in SMPL-X parameters cannot effectively supervise the orientation of the generated human mesh. Thus, we define the orientation of the human mesh through the joint points for better human mesh orientation supervision (see Figure 4). Specifically, a human's left hip $\hat{J}_{lhip}$ and right hip $\hat{J}_{rhip}$ can provide the direction of the x-axis, and a human's pelvis $\hat{J}_{pelvis}$ and spine $\hat{J}_{spine}$ can offer the direction of the y-axis. We use the cross product of the x-axis vector and the y-axis vector to obtain the ground truth orientation $\hat{\mathcal{O}}$ of the human body: $\hat{\mathcal{O}} = (\hat{J}_{lhip} - \hat{J}_{rhip}) \times (\hat{J}_{spine} - \hat{J}_{pelvis})$. In this way, we compute the orientation loss $\mathcal{L}_{\mathcal{O}}$ between the prediction joints $\mathcal{O}$ and ground-truth joints $\hat{\mathcal{O}}$ as: $\mathcal{L}_{\mathcal{O}} = |\mathcal{O} - \hat{\mathcal{O}}|$.

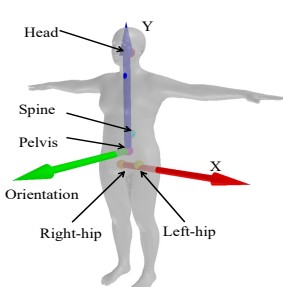

Figure 4: The orientation (green arrow) defined from human joints.

**3D joint density loss**. We use 3D Gaussian kernels to generate a density map of 3D joints from GT $\hat{J}_{3D}$ and prediction $J_{3D}$. Unlike the direct $L_1$ loss of 3D joint locations (as in mesh loss), we use mean square error loss (MSE) for the 3D density map regression:

$$\mathcal{L}_{denj3d} = ||\mathbf{Gau}(J_{3D}) - \mathbf{Gau}(\hat{J}_{3D})||_2^2, \quad (2)$$

where **Gau** stands for the Gaussian smoothing step, which generates a 3D Gaussian probability map centered around the joint locations. The 3D joint density loss $\mathcal{L}_{denj3d}$ is conducted elemental-wisely in 3D space and provides stronger supervision for the pose of the human mesh, handling the pose ambiguities better in the MVMP HMR task under occlusions.

In total, the whole training loss is: $\mathcal{L} = \mathcal{L}_D + \lambda_2 \mathcal{L}_P + \mathcal{L}_{mesh} + \lambda_3 \mathcal{L}_{\mathcal{O}} + \lambda_4 \mathcal{L}_{denj3d}$. We set $\lambda_2 = 10$, $\lambda_3 = 5$ and $\lambda_4 = 1$ in our experiments.

## 4 EXPERIMENTS AND RESULTS

### 4.1 DATASET

We perform the experiments on 3 datasets: MVMP-HMR collected by us, Panoptic (Joo et al., 2015b), and Human3.6M (Ionescu et al., 2013). The collection process of MVMP-HMR is as follows.

Table 2: The result comparison on our MVMP-HMR, Human3.6M and CMU Panoptic dataset. Rows 1-6 are single-view HMR SOTAs with multi-view fusion techniques, Rows 7 is multi-view single person HMR SOTAS, and Rows 8-9 are 3D pose estimation methods modified for SMPL-X regression. In the table, the best results are highlighted in **bold**, while the second-best results are underlined.

| Dataset | MVMP-HMR | | | Human3.6M | | | CMU Panoptic | | |
|---|---|---|---|---|---|---|---|---|---|
| Method | MPJPE ↓ | PVE ↓ | PA-PVE ↓ | MPJPE ↓ | PVE ↓ | PA-PVE ↓ | MPJPE ↓ | PVE ↓ | PA-PVE ↓ |
| 3DCrowdNet (Dist) | 221.2 | 284.3 | 72.2 | 135.8 | 130.9 | 69.5 | 443.2 | 456.3 | 186.7 |
| AiOS (Dist) | 873.6 | 642.4 | 110.5 | 156.8 | 133.4 | 78.9 | 730.6 | 550.9 | 195.8 |
| TokenHMR (Dist) | 632.3 | 661.3 | 191.5 | 112.4 | 122.5 | 58.9 | 616.3 | 598.0 | 194.4 |
| Multi-HMR (Dist) | 841.0 | 651.4 | 71.0 | 98.5 | 97.3 | 46.3 | 568.7 | 453.4 | 195.1 |
| Multi-HMR (Avg) | 752.5 | 753.6 | 61.7 | 110.3 | 99.8 | 52.7 | 546.9 | 509.8 | 220.8 |
| Multi-HMR (Fusion) | 602.4 | 529.5 | 111.4 | 129.7 | 122.8 | 65.4 | 523.3 | 423.1 | 192.6 |
| HeatFormer | 185.5 | 148.3 | 83.6 | **60.3** | **65.4** | **31.2** | 385.6 | 376.2 | 125.6 |
| VoxelSMPLX (Only) | 225.4 | 262.0 | 240.6 | 147.5 | 160.3 | 54.5 | 372.1 | 365.5 | 134.2 |
| VoxelSMPLX (Joint) | 288.6 | 427.4 | 317.1 | 156.3 | 167.2 | 61.3 | 403.5 | 385.9 | 158.6 |
| MVMP-HMR (Ours) | **177.5** | **129.2** | **51.8** | 93.5 | 92.1 | 44.3 | **278.6** | **234.5** | **95.3** |

**Dataset Generation**. To study multiview multi-person human mesh recovery (HMR), we introduce MVMP-HMR, a large-scale dataset generated using the virtual game platform GTA-V. The dataset features diverse everyday scenes (e.g., basketball courts, factories, streets) with varying numbers of people (10–30 per scene), complex occlusions, and up to 50 camera views per scene. Using GTA-VAPIs, we extract 98 3D body keypoints, depth maps, and semantic masks for each scene. In total, MVMP-HMR contains 15 complex scenes, making it the first large-scale multiview multi-person HMR dataset, designed to advance HMR research in challenging, real-world-like environments.

**Dataset Annotation**. Since GTA-V APIs do not provide 3D mesh labels, we adopt an HMR method (Baradel et al., 2024) for SMPL-X annotation in 3D world coordinates. To obtain accurate SMPL-X parameters, we first apply (Baradel et al., 2024) on all views of a frame to obtain SMPL-X labels in the camera coordinates of all people. *Then, for each person, we match the ground-truth 2D keypoints provided in GTA-V and the ones extracted from the predicted SMPL-X labels of all views. The SMPL-X label with the lowest matching error is assigned as the ground truth of the corresponding person*. In contrast to the single-view HMR task, the MVMP-HMR task estimates the human meshes in 3D world coordinates. Thus, we transform these 'predicted' ground-truth human meshes to world coordinates via a rotation and translation matrix.

From the single-view HMR prediction, we obtain global orientation $R_{cam}$ and translation $T_{cam}$ to decide the directions and locations of the human mesh in camera coordinates. We then compute $R, T$ between 3D joint points shared in predicted SMPL-X mesh format (camera coordinates) and GTA-V (world coordinates). Then ground truth (GT) global orientation parameter $R_{wld}^{gt}$ and translation parameter $T_{wld}^{gt}$ are formulated as: $R_{wld}^{gt} = R \cdot R_{cam}$ and $T_{wld}^{gt} = T_{cam} + T$. The SMPLX annotation acquisition for the real dataset Panoptic (Joo et al., 2015b) is consistent with the above content. SMPLX label in Human3.6m are obtained from Choi et al. (2022b)

### 4.2 EXPERIMENT SETTINGS

**Implementation**. In experiments, we divide the 15 scenes in the dataset according to the distribution of people numbers, and the ratio of the training/testing set is 2:1. We use VIT-L (Dosovitskiy et al., 2021) as our model feature extraction backbone. We pre-train the posenet (Sun et al., 2019) and rootnet (Tu et al., 2020) for 60 epochs on our dataset for detection. The input images are resized to 1288 x 1288 with zero paddings. We adopt Adam as the optimizer with 5e-5 learning rate. The training epoch is 50, and the training is conducted on 2 RTX6000 Ada GPUs, with a batch size of 1.

**Comparison methods.** We compare our MVMP-HMR method with multi-person HMR SOTAs with multiview settings and 3D HPE method for HMR tasks. Single-view HMR SOTAs Multi-HMR (Baradel et al., 2024), 3DCrowdNet (Choi et al., 2022b), AiOS (Sun et al., 2024), and TokenHMR (Dwivedi et al., 2024) first conduct predictions of each view, then use a multi-view matching algorithm to match the prediction results of each person under multiple views, and fuse the prediction results of each person in the scene under multiple views into the final result. The fusion strategy includes selecting the closest one as the prediction result based on the distance from the camera (denoted as

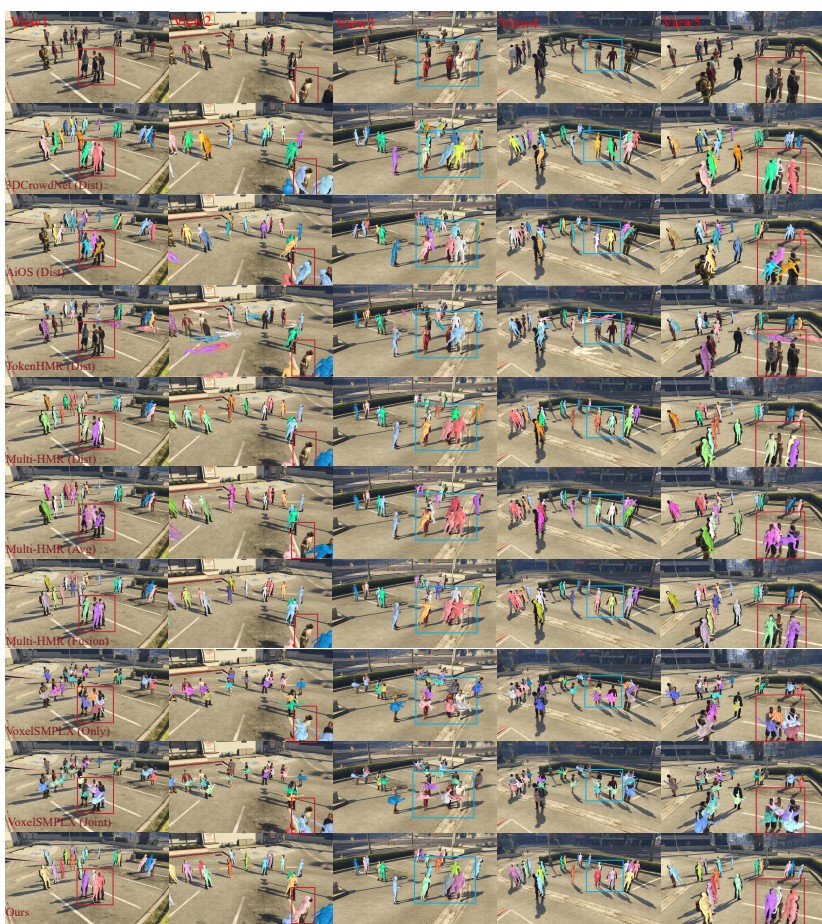

Figure 5: The top row is the multiview input, and each subsequent row is the 3D predictions of the methods projected to view plane. Red boxes indicate that our method can better handle occlusions than comparison methods. Blue boxes indicate our method achieves better posture than comparisons.

Table 3: Loss term ablation study. The first row does not use any new loss, the second row only adds the orientation loss, the third row only adds the 3D joint density loss, and the last row adds both new losses (our method).

| Loss | MPJPE ↓ | PVE ↓ | PA-PVE ↓ |
|---|---|---|---|
| $\mathcal{L}_D + \lambda_2 \mathcal{L}_P + \mathcal{L}_{mesh}$ | 217.1 | 161.7 | 120.4 |
| $+5.0\mathcal{L}_\mathcal{O}$ | 187.9 | 144.8 | 89.0 |
| $+1.0\mathcal{L}_{denj3d}$ | 180.7 | 132.4 | 50.2 |
| +Both (Ours) | **177.5** | **129.2** | **51.8** |

'Dist'), using an average strategy to fuse the results of each view prediction (denoted as 'Avg'), and using a sub-network to predict the weight value corresponding to each view prediction to fuse the final result (denoted as 'Fusion'). We also compare with a multi-view 3D pose estimation method VoxelPose (Tu et al., 2020). We sample human queries from the feature volume with the predicted joint locations of VoxelPose (Tu et al., 2020) and then estimate the SMPL-X parameters from the human queries with regression MLPs. There are two variants: use the pretrained VoxelPose and only train the regression MLPs, denoted 'VoxelSMPLX (Only)'; or jointly train VoxelPose and MLPs, denoted as 'VoxelSMPLX (Joint)'. We extend the single-person HeatFormer (Matsubara & Nishino, 2025) to multi-person scenarios via a top-down framework, decomposing the scene into individual instances for independent SMPL-X regression.

### 4.3 MVMP HMR RESULTS

We comprehensively evaluate our MVMP-HMR model against state-of-the-art approaches on three benchmarks: MVMP-HMR (synthetic), Human3.6M, and the CMU Panoptic Dataset, as shown

Table 4: Feature fusion method ablation study.

| Fusion Method | MPJPE ↓ | PVE ↓ | PA-PVE ↓ |
|---|---|---|---|
| Deformable | 261.3 | 207.2 | 80.6 |
| Max | 245.2 | 193.5 | 74.8 |
| Mean (Ours) | **177.5** | **129.2** | **51.8** |

Table 5: Primary joint ablation study.

| Primary Joint | MPJPE ↓ | PVE ↓ | PA-PVE ↓ |
|---|---|---|---|
| Head | 280.6 | 172.3 | 68.2 |
| Spine | 190.2 | 146.9 | 86.1 |
| Pelvis (Ours) | **177.5** | **129.2** | **51.8** |

in Table 2. The comparison includes six single-view HMR baselines equipped with multi-view fusion techniques (3DCrowdNet, AiOS, TokenHMR, Multi-HMR variants) and a 3D HPE method with SMPL-X regression (VoxelSMPLX). We further compare with HeatFormer, a recent multi-view transformer-based framework originally designed for single-person mesh recovery. Overall, MVMP-HMR achieves consistently superior performance over all multi-person competitors across the three benchmarks. Existing comparison methods are generally built for either single-view HMR or 3D HPE in simple scenes, making them unable to robustly integrate multi-view cues or recover accurate meshes solely from pose estimations. In contrast, our approach specifically targets the more challenging multi-view multi-person setting, effectively handling severe inter-person occlusions and depth ambiguities. Although HeatFormer reports the best results on Human3.6M, this is expected because Human3.6M contains only one subject per scene, aligning with HeatFormer's single-person design. This demonstrates the advantages of the proposed MVMP-HMR model in handling severe occlusions and human orientation or pose ambiguities in complex scenes.

As **visualized** in Figure 5, our proposed method outperforms all comparison methods, in terms of predicting completeness (no person is missed) and pose accuracy. The *red boxes* indicate our method can handle occlusions well and estimate meshes accurately for occluded persons, while all comparisons neglect the occluded persons or produce wrong shapes. The *blue boxes* indicate our method achieves more natural and realistic human poses, with better limb positioning and alignment compared to comparison methods that produce unrealistic limb orientations and poses (such as flying pose in the first row, fourth column of 3DCrowdNet (Dist), hugging posture in the six row, third column of Multi-HMR (Fusion), or VoxelSMPLX).

## 4.4 QUALITATIVE RESULTS ON REAL WORLD DATASET

To intuitively evaluate the effectiveness of our proposed framework, we visualize the reconstruction results on two distinct benchmarks: the Human3.6M dataset and the CMU Panoptic dataset. The visualization results verify that our method generalizes well from single-person distinct poses to complex multi-person interactions.

**Human3.6M.** As shown in the bottom rows of Figure 6, our model produces accurate and consistent 3D meshes across different viewpoints. It captures fine-grained pose details and maintains precise alignment with image evidence, even in cases with rapid motion or self-occlusion (e.g., sitting, crouching). The stable performance across diverse actions confirms the robustness of our single-person reconstruction.

**CMU Panoptic.** To evaluate performance in more challenging settings, we train and visualize results on the CMU Panoptic dataset, which contains multiple closely interacting subjects and severe occlusions. As illustrated in Figure 7, our approach remains robust in these crowded scenes, successfully separating inter-person cues and reconstructing accurate meshes for all individuals, including in highly occluded sequences such as "Pizza" and "Band." This demonstrates that our method scales effectively to real-world multi-person scenarios and can be deployed in practical, in-the-wild applications.

## 4.5 ABLATION STUDY

**Loss term ablation study.** We conduct ablation studies on two novel losses—orientation Loss $\mathcal{L}_{\mathcal{O}}$ and 3D joint density Loss $\mathcal{L}_{denj3d}$ —by incorporating them individually or together with three standard single-view HMR losses. As shown in Table 3, both new losses improve the performance of our MVMP-HMR model, and using both together achieves the best results, demonstrating their effectiveness in reducing orientation and pose ambiguities in multiview multi-person HMR. Notably,

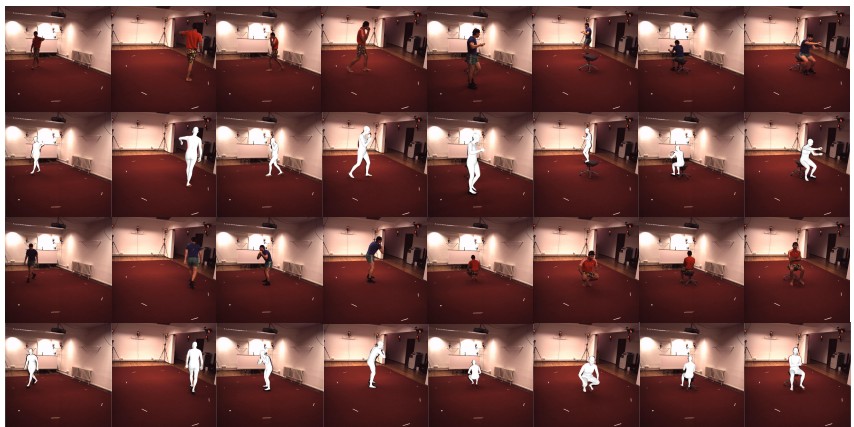

Figure 6: Qualitative results on on the Human3.6M dataset.

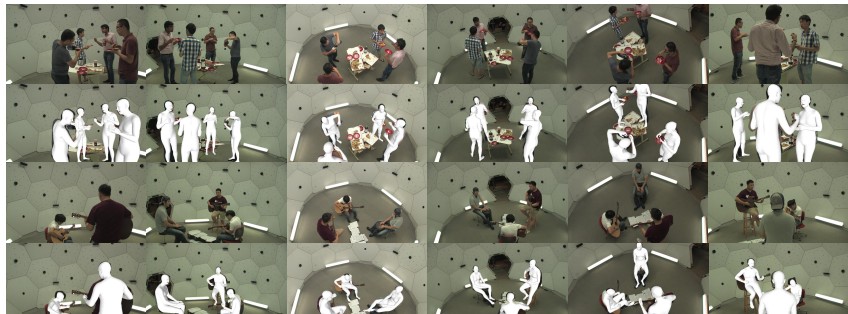

Figure 7: Qualitative results on the CMU Panoptic dataset.

$\mathcal{L}_{denj3d}$ contributes more significantly, providing stronger 3D supervision and greater performance gains. **See detailed loss term weight ablations in Table 9 of the Appendix.**

**Feature fusion method ablation study.** We also perform ablation studies on the feature fusion method, using three different methods: Deformable attention, Max, and Mean. As in Table 4, the performance using the mean operation to fusion multi-view features achieves marginally superior performance than using deformable attention or max. The possible reason is that the mean method is simple and efficient, suitable for global information fusion, but max is suitable for highlighting key features, but is susceptible to noise interference. And the deformable attention has a high computational overhead. In our setting, the mean operation is better for our environment to aggregate multi-view features. Thus, in our experiments, we use the mean as the feature fusion method.

**Primary joint selection ablation study.** To determine the optimal primary joint for our model, we conducted an ablation study comparing three different primary joints: the pelvis, head, and spine. As in Table 5, the results show that the use of the pelvis for localisation produces marginally better performance. This can be attributed to the pelvis's stability across various viewpoints and its central location, which allows for more complete human body information to be captured in the model's queries. Consequently, we chose the pelvis as the primary joint for all subsequent experiments. **See model architecture and view number ablations in the Appendix**.

## 5 CONCLUSION

In this paper, we propose a novel multi-person whole-body human mesh recovery model from multiview images and a new large multiview HMR benchmark with more persons in large occluded scenes. As far as we know, this is the first study on multiview-multiperson-based (MVMP) HMR tasks and the first large MVMP-HMR benchmark in this area. Besides, two novel losses are put forward to further enhance the model's performance: the orientation loss and the 3D joint density loss, handling the orientation and pose ambiguities in the mesh predictions under the occluded scenes. The experiments validate that the MVMP-HMR model can deal with the occlusion issue better than existing single-view HMR SOTAs. The proposed model and benchmark shall extend the HMR task to more complicated scenes with wider application scenarios.

## ETHICS STATEMENT

This work introduces a framework for multiview multi-person human mesh recovery (MVMP-HMR) using a synthetic dataset generated with the GTA-V engine and publicly available benchmarks such as Human3.6M (Ionescu et al., 2013) and CMU Panoptic (Joo et al., 2015b), all of which contain no personally identifiable information. SMPL-X annotations are derived automatically using existing HMR models, reducing the need for manual labeling and associated privacy concerns. Our research advances human mesh recovery with potential benefits in motion analysis, human-computer interaction, and safety-critical applications. While we are not aware of negative societal impacts specific to our method, we acknowledge broader ethical considerations related to surveillance, fairness, and potential misuse, and emphasize responsible and transparent deployment.

## REPRODUCIBILITY STATEMENT

We provide detailed descriptions of our MVMP-HMR architecture, including the ViT-L (Dosovitskiy et al., 2021) backbone, multi-view feature fusion, and Human Transformer Block, along with the proposed orientation and 3D joint density losses. Implementation details such as training configuration, hyperparameters, and dataset splits are reported in Section 4.2. Experiments are conducted on 2 NVIDIA RTX 6000 Ada GPUs, and we will release the MVMP-HMR dataset, source code, pre-trained models, and training logs upon acceptance, ensuring reproducibility and facilitating future research.

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

# A APPENDIX

## A.1 METRIC DETAILS.

We evaluate the HMR predictions with metrics MPJPE, PVE, and PA-PVE, but **in 3D space**, not in camera view as previous single-view HMR tasks.

- *MPJPE*: Mean Per Joint Position Error measures the average Euclidean distance between predicted 3D joints and ground truth 3D joints.
- *PVE*: Mean Per-vertex Error is defined as the average point-to-point Euclidean distance between predicted mesh vertices and ground truth mesh vertices. It is proposed to calculate in the world space.
- *PA-PVE*: Procrustes-aligned PVE is calculated according to PVE after executing Procrustes Analysis to align predicted mesh vertices with ground truth mesh vertices.

MPJPE and PVE are the main metrics in our task. All our keypoints and vertices are obtained from the corresponding SMPL-X models through SMPL-X parameters, and the unit of all our metrics are mm in world coordinates.

## A.2 MAIN FEATURES OF MVMP-HMR VS SOTA HMR AND HPE METHODS.

As shown in Table 6, we have compared over 10 human mesh recovery and human pose estimation methods. It is easy to see that our method is the only one that focuses on the multiview multi-person human mesh recovery task.

Table 6: Comparison of existing HMR and HPE Methods. None of them meets our setting without revision.

| Method | Multi-view | Multi-person | 3D Pose | Mesh |
|---|---|---|---|---|
| HMR(Kanazawa et al., 2018) | × | × | ✓ | ✓ |
| PyMAF-X(Zhang et al., 2021a) | × | × | ✓ | ✓ |
| OSX(Lin et al., 2023) | × | × | ✓ | ✓ |
| SMPLer-X(Cai et al., 2023) | × | × | ✓ | ✓ |
| 3D CrowdNet(Choi et al., 2022b) | × | ✓ | ✓ | ✓ |
| AiOS(Sun et al., 2024) | × | ✓ | ✓ | ✓ |
| TokenHMR(Dwivedi et al., 2024) | × | ✓ | ✓ | ✓ |
| Multi-HMR(Baradel et al., 2024) | × | ✓ | ✓ | ✓ |
| U-HMR(Yu et al., 2022) | ✓ | × | ✓ | ✓ |
| HeatFormer(Matsubara & Nishino, 2025) | ✓ | × | ✓ | ✓ |
| VoxelPose(Tu et al., 2020) | ✓ | ✓ | ✓ | × |
| Faster VoxelPose(Ye et al., 2022) | ✓ | ✓ | ✓ | × |
| MVP(Wang et al., 2021) | ✓ | ✓ | ✓ | × |
| MVMP-HMR (Ours) | ✓ | ✓ | ✓ | ✓ |

Note: ✓ indicates supported, × indicates not supported.

## A.3 DATASET

The GTA-V game engine demonstrates exceptional authenticity and has been widely adopted for dataset generation across various research fields, including GTA-Human (Cai et al., 2024b) and the multi-view counting dataset CVCS(Zhang et al., 2021b), offering highly realistic scenes, dynamic weather systems, comprehensive lighting variations, and diverse human activities such as walking, phone usage, drinking, smoking, listening to music, and social interactions. Our dataset shows a strong bias toward clear/sunny conditions (78.12%) with overcast coverage (12.78%) and adverse weather (9.09%), while temporal distribution exhibits pronounced daytime bias (79.73% between 6:00-18:00) with activity peaks during commuting hours and sparse nighttime coverage (20.27%). Compared to traditional 3D HPE datasets that are primarily collected in controlled laboratory settings, our GTA-V-generated dataset focuses on outdoor practical application scenarios with broader scene diversity

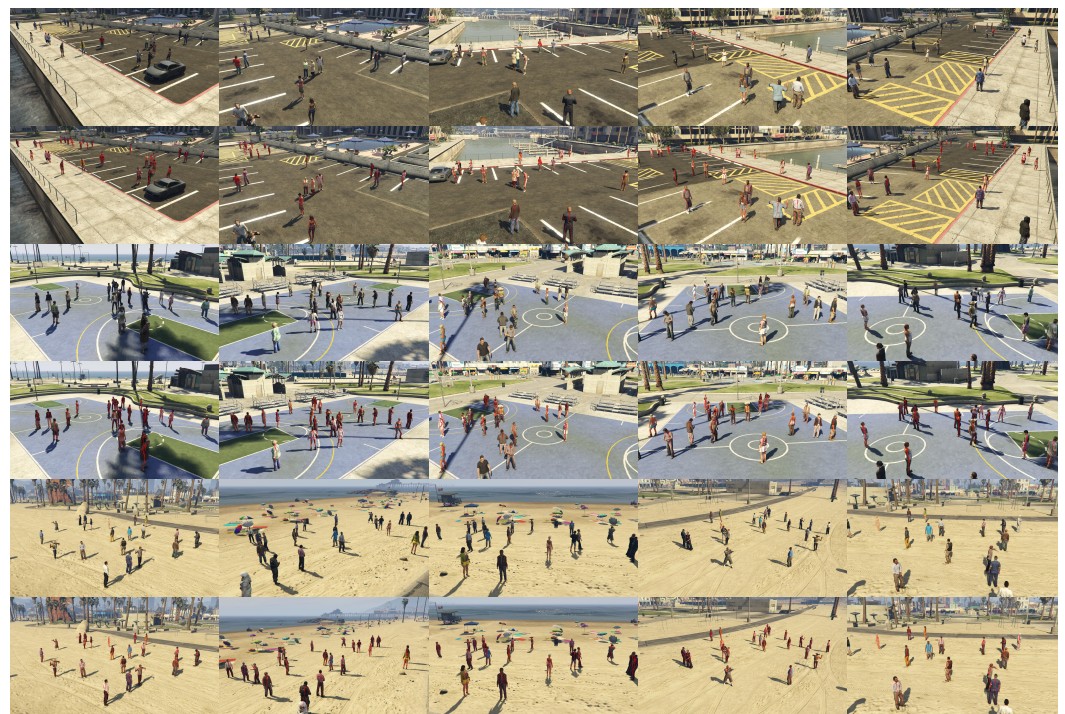

Figure 8: The visualization examples of the other scenes in the dataset. Red joints mean the keypoints of humans. The red line means the skeleton of people.

and enhanced ecological validity, better representing the complexity and variability encountered in real-world conditions. More detailed dataset analysis to be added to the paper.

## A.4 MORE VISUALIZATIONS OF OUR MVMP-HMR DATASET

We have introduced the dataset MVMP-HMR in the main text. Now we will show some other scenes in our dataset with their cooperation 3D joints, which are also key points for our dataset annotation. Figure 8 shows three scenes in our dataset. We can see the details of the 2D key-points location with red color. In our setting, the one who can't be seen completely at this view, their keypoint location will be dropped. These 2D keypoints are all projected from 3D keypoints. We can also provide precise keypoint locations for the multiview pose estimation task.

## A.5 QUALITIVE COMPARISION RESULTS ON HUMAN3.6M

We first compare our method against existing state-of-the-art approaches on the Human3.6M dataset. As illustrated in Figure 9, our method generates high-fidelity human meshes that align precisely with the input images. Hand and Arm Reconstruction (Yellow, Red, and Green Boxes): As shown in the cam1 (Yellow), cam3 (Red), and cam4 (Green) rows, the baseline methods often fail to capture the precise articulation of the arms and hands, leading to noticeable deviations from the image evidence. In contrast, our method (Column 2) accurately recovers the 3D pose of the upper limbs, maintaining strict alignment with the input images even during complex gestures. Foot and Leg Alignment (Blue Box): The cam2 row (Blue) highlights the lower body during a walking motion. While competing methods exhibit "sliding" artifacts or incorrect knee bending angles, our approach ensures precise foot-ground contact and leg orientation, demonstrating the effectiveness of our multi-view feature fusion in resolving depth ambiguities.

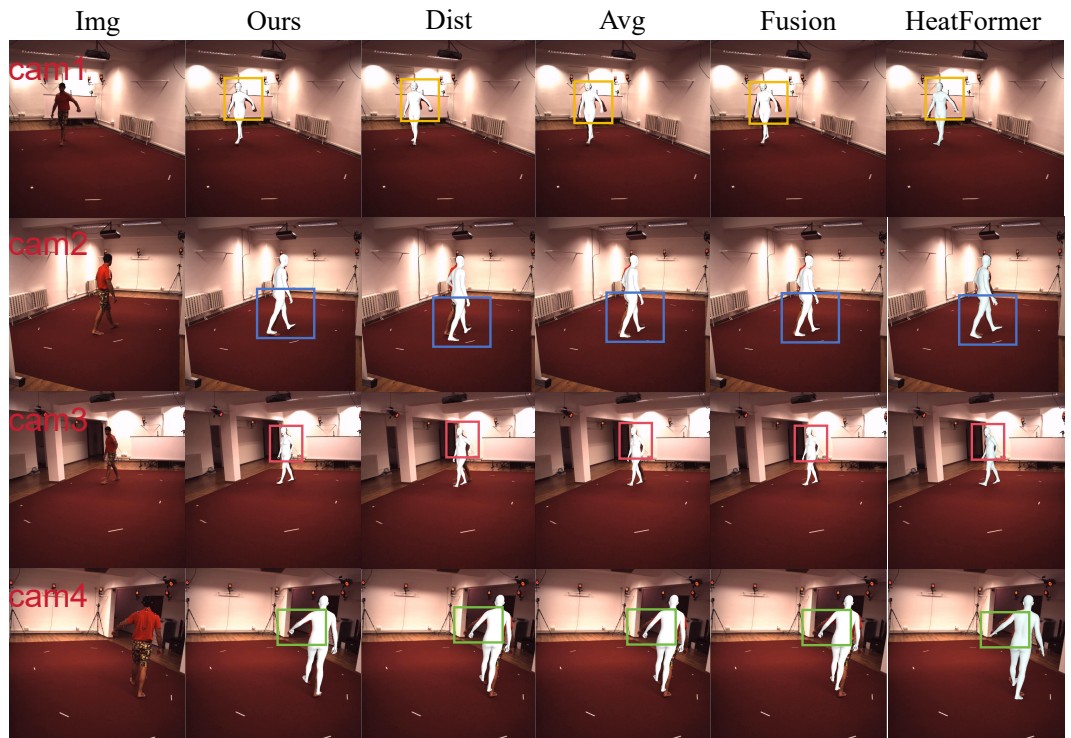

Figure 9: Qualitative comparison with state-of-the-art methods on the Human3.6M dataset.

## A.6 MODEL BRANCH ABLATION STUDY

In our method, we adopted a dual-branch structure, where one branch is used for feature extraction for SMPL-X parameter regression, and the other branch is used for pose heatmap prediction to assist positioning. In addition, we have changed the dual-branch structure to a single-branch structure as an ablation study: the features extracted by the single branch are followed by two task heads for SMPL-X parameter regression and heatmap prediction. The experiments in Table 7 show that the results of using a single branch are not as good as those of the dual branch. Since the detection branch in the dual branch is pre-trained by us, mixing SMPL-X parameter regression and detection together will cause inaccurate detection results and affect our parameter regression. We can see that the results reflect the dual-branch method's advantages compared to the single-branch method.

Table 7: The experiment results of the single-branch and the dual-branch of our proposed method.

| Method | Precision ↑ | Recall ↑ | F1-Score ↑ | MPJPE ↓ | PVE ↓ | PA-PVE ↓ |
|---|---|---|---|---|---|---|
| Single-Branch | 90.0 | 82.0 | 86.0 | 204.3 | 164.9 | 117.1 |
| Dual-Branch | **99.0** | **89.0** | **94.0** | **177.5** | **129.2** | **51.8** |

## A.7 MODEL PARAMETERS AND INFERENCE TIME

In addition to the results displayed in the dataset compared with other methods, we also made comparisons regarding model parameters and inference speed in Table 8. Our model parameters only count the model parameters during testing. Our inference time calculation is to run the model for 100 sample inputs and then test the entire test set for an average test time. From our model framework, it can be seen that the 3D voxel features constructed from multi-view feature projections and fusion, as well as the subsequent network processing, are very resource-intensive. However, our model's parameters and inference speed achieve a moderate result compared to single-view HMR and multi-view HPE methods. Although the HPE method has a simpler network architecture, resulting in lower estimated model parameters and inference speed than ours, the HPE method can't achieve good

Table 8: The model parameters and inference time compared to HMR and HPE SOTAs.

| Method | Model Parameters (MB) ↓ | Inference Time (s) ↓ |
|---|---|---|
| 3DCrowdNet (Dist) (Choi et al., 2022b) | 931.92 | 1.12 |
| AiOS (Dist) (Sun et al., 2024) | 1122.28 | 0.97 |
| TokenHMR (Dist) (Dwivedi et al., 2024) | 2598.57 | 2.44 |
| Multi-HMR (Dist) (Baradel et al., 2024) | 1210.17 | 2.33 |
| Multi-HMR (Avg) (Baradel et al., 2024) | 1210.17 | 2.33 |
| Multi-HMR (Fusion) (Baradel et al., 2024) | 1331.19 | 2.53 |
| VoxelSMPLX (Only) (Tu et al., 2020) | 404.45 | 1.00 |
| VoxelSMPLX (Joint) (Tu et al., 2020) | 404.45 | 1.00 |
| MVMP-HMR (Ours) | 1380.28 | 1.59 |

Table 9: Loss term weight ablation study. The first row does not use any new loss. Rows 2-5 add the orientation loss, and Rows 6-11 add both the orientation and 3D joint density loss.

| Loss | MPJPE ↓ | PVE ↓ | PA-PVE ↓ |
|---|---|---|---|
| $\mathcal{L}_D + \lambda_2 \mathcal{L}_P + \mathcal{L}_{mesh}$ | 217.1 | 161.7 | 120.4 |
| $+2.0 * \mathcal{L}_{\mathcal{O}}$ | 201.6 | 151.9 | 99.4 |
| $\mathbf{+5.0} * \mathcal{L}_{\mathcal{O}}$ | 187.9 | 144.8 | 89.0 |
| $+10.0 * \mathcal{L}_{\mathcal{O}}$ | 195.1 | 149.2 | 80.7 |
| $+100.0 * \mathcal{L}_{\mathcal{O}}$ | 195.0 | 150.0 | 71.8 |
| $+5.0 * \mathcal{L}_{\mathcal{O}} + 0.1 * \mathcal{L}_{denj3d}$ | 190.2 | 144.5 | 89.4 |
| $+5.0 * \mathcal{L}_{\mathcal{O}} + 0.2 * \mathcal{L}_{denj3d}$ | 190.7 | 149.7 | 83.3 |
| $+5.0 * \mathcal{L}_{\mathcal{O}} + 0.5 * \mathcal{L}_{denj3d}$ | 187.6 | 147.4 | 87.9 |
| $\mathbf{+5.0} * \mathcal{L}_{\mathcal{O}} + \mathbf{1.0} * \mathcal{L}_{denj3d}$ (**Ours**) | **177.5** | **129.2** | **51.8** |
| $+5.0 * \mathcal{L}_{\mathcal{O}} + 2.0 * \mathcal{L}_{denj3d}$ | 293.3 | 149.5 | 69.6 |
| $+5.0 * \mathcal{L}_{\mathcal{O}} + 5.0 * \mathcal{L}_{denj3d}$ | 368.8 | 160.0 | 69.4 |

results on our MVMP-HMR dataset. Single-view HMR does not involve the fusion of multi-view features, so its model parameter count is smaller than ours. Additionally, the efficiency of detecting directly on 3D voxel features is higher than that of multi-view matching, leading to shorter inference times for our method.

## A.8 LOSS TERM WEIGHT ABLATION STUDY.

We conduct the loss term weight ablations for the proposed orientation loss ($\mathcal{L}_{\mathcal{O}}$) and 3D joint density loss ($\mathcal{L}_{denj3d}$) in Table 9. The first row uses the loss usually used in prior work (Baradel et al., 2024). Row 2-5 add the proposed orientation loss $\mathcal{L}_{\mathcal{O}}$ with different $\lambda_3$ weights, and the performance all improved compared to without it, demonstrating the effectiveness of the $\mathcal{L}_{\mathcal{O}}$ loss. $\lambda_3 = 5.0$ achieves the best results, and we use it as the loss weight of $\mathcal{L}_{\mathcal{O}}$ in the experiments. Row 6-11 further add the proposed 3D joint density loss $\mathcal{L}_{denj3d}$ in the model training. $\lambda_3 = 5.0, \lambda_4 = 1.0$ achieves the best results. When $\lambda_4$ is too large, the 3D joint density loss may decrease the human mesh prediction performance because $\mathcal{L}_{denj3d}$ might be too strong.

Table 10: The backbone ablation study and using ViT-L is the best

| Backbone | MPJPE ↓ | PVE ↓ | PA-PVE ↓ |
|---|---|---|---|
| ViT-S (Dosovitskiy et al., 2021) | 201.6 | 157.8 | 64.9 |
| ViT-B (Dosovitskiy et al., 2021) | 185.7 | 141.8 | 61.6 |
| ViT-L (Dosovitskiy et al., 2021) | **177.5** | **129.2** | **51.8** |

## A.9 FEATURE EXTRACTION BACKBONE MODEL

We also perform ablation studies on the feature extraction backbone models, using three different feature extraction backbone models: ViT-S, ViT-B, and ViT-L (Dosovitskiy et al., 2021), differing in model sizes: small, base, and large. As in Table 10, the result of using ViT-L as the backbone model

Table 11: Testing camera view number ablation study: the model is trained on 5 views and tested with 3-20 views.

| ViewNum | MPJPE ↓ | PVE ↓ | PA-PVE ↓ |
|---|---|---|---|
| 3 | 193.6 | 137.6 | 50.9 |
| 5 | 177.5 | 129.2 | 51.8 |
| 7 | 171.0 | 125.2 | 48.2 |
| 9 | 168.1 | 122.0 | 47.9 |
| 15 | 166.9 | 120.3 | 45.5 |
| 20 | **164.8** | **118.4** | **44.3** |

is the best, which has more model parameters with stronger feature extraction ability. Therefore, we use ViT-L as the feature backbone model in our MVMP-HMR model.

## A.10 TESTING VIEW NUMBER ABLATION STUDY

Finally, we perform ablation studies on the input camera view number in the testing stage. The model is trained with 5 camera views and tested with different camera views, ranging from 3-20 camera views, shown in Table 11. We observe that as the testing camera view number increases, the model's performance also improves. The reason is that with more camera views, more clues are provided, and the proposed Multiview-HMR model can effectively fuse multiview information to handle the occlusions in the scene. The model performance change is not quite large when the camera view number increases, also indicating our model's robustness to different view numbers. In addition, this experiment also demonstrates that our model has good generalization ability in terms of the number of viewpoints.

## A.11 LIMITATIONS

In our experimental setting, we require the input to be multiple cameras that have been calibrated to obtain the internal and external parameters of the camera. Although this is difficult to obtain in the real world, many existing excellent multi-view matching algorithms (such as (Schönberger et al., 2016)) or VGGT (Wang et al., 2025) can perform camera calibration through multiple perspectives, which provides great help for the future application of our method. In the future, we can consider how to use multi-view without camera parameter calibration to perform multiview multi-person human mesh recovery.