# OpenReview forum: "MVMP-HMR: Multiview Multi-Person Human Mesh Recovery Under Large Scenes with Occlusions"
_ICLR.cc/2026/Conference — Submitted to ICLR 2026_

### Official Review · Reviewer_VktQ · 2025-10-30

**Soundness:** 2
**Presentation:** 3
**Contribution:** 2
**Rating:** 4
**Confidence:** 4

**Summary:**

This paper presents an approach for multi-person whole-body human mesh recovery model from multiview images and introduce a multiview HMR benchmark with multiple persons in large occluded scenes. The key idea is to first fuses multiple views to obtain
a 3D feature volume for all persons, and then the pelvis joint from a 3D pose estimation net is utilized to acquire the human query of each person from the 3D feature volume. Finally, the human queries are cross-attentioned with the 3D feature volume and integrated to decode each person’s 3D meshes. A orientation loss and a 3D joint density loss is developed for training.

**Strengths:**

This work introduces multiview multi-person HMR task under large scenes with severe occlusions, and accordingly develop a method for MVMP-HMR.

A large MVMP-HMR dataset is introduced for training and evaluation.

Experiments demonstrate the superiority of the proposed method.

The paper is well-written and easy to read.

**Weaknesses:**

The paper is submitted to the track of 'datasets and benchmarks', while the authors highlight the method throughout the paper without introducing and analyzing the dataset in detail (including factors such as scene complexity, diversity, occlusion ratio, number of views, count of person, etc).

In my opinion, the proposed method actually has very limtied novelty. Most technical components are off-the-shelf. The orientation loss is devised because the collected dataset has the correpsonding supervision, with very limited contribution.

The experiments are not convincing. First of all, most compared methods are tailored for single-view HMR, rather than multi-view HMR. Besides, some recent multi-view HMR methods are not compared, e.g., HeatFormer.

The dataset is contructed with the HMR method of Baradel et al. 2024. Therefore, a direct comparison with the method is necessary.

I doubt how the proposed method generalizes to unseen data, as the collected data contains only about 60K frames covering 15 scenes from GTA-V.

The paper lacks visual ablation studies for validation.

Discussion on the computational complexity and efficiency should be included.

**Questions:**

How does the number of views affect the performance of the proposed method?

---

> ### Author Response · Authors · 2025-11-28
> **Response to Reviewer VktQ**
>
> Thank you for your valuable suggestions. Each question is answered as follows. We hope the response has addressed your concerns. We're glad to have further communication with you. The revision in the paper is marked in blue.
>
> **Q1: The paper is submitted to the track of 'datasets and benchmarks', while the authors highlight the method throughout the paper without introducing and analyzing the dataset in detail (including factors such as scene complexity, diversity, occlusion ratio, number of views, count of person, etc).**
>
> The paper proposes both a novel method and a benchmark dataset. We provide detailed dataset statistics in **Table 1** (Area, SceneNum, Subjects, Occlusion level, Views, Frames), comparing it favorably to existing datasets like BEDLAM and AGORA. In **Lines 800-837**, we provide some detailed descriptions  (weather, lighting) of the dataset. In addition, we also compare it with datasets in the field in the main text, and we provide our own baseline method.
>
> **Q2: In my opinion, the proposed method actually has very limtited novelty. Most technical components are off-the-shelf. The orientation loss is devised because the collected dataset has the correpsonding supervision, with very limited contribution.**
> Our task--multi-view multi-person human mesh recovery--is also a multi-view based task, sharing similarities in model architectures with other multi-view based tasks, such as multi-view 3D reconstruction, multi-view 3D pose estimation, or multi-view counting/detection/tracking, etc., generally using a single-view feature extraction, multi-view fusion, and multi-view prediction fashion. However, compared to these existing multi-view task models, we still have major differences:
>
> 1) Our task is different--we propose an end-to-end multi-view model for the multi-view multi-person HMR task, which adopts a different decoder head for the task.
> 2) We propose to apply a localization network to assist the HMR tasks for multiple persons in the 3D world, which improves the model efficiency and effectiveness. To verify this, we conducted an extra experiment by removing the localization network branch, whose results are as follows.It demonstrates that our dual-branch approach outperforms the single-branch approach in prediction, helping us to locate more accurately and obtain more precise human features. We have added the results to the paper.
>
>
> | Method        | Precision ↑ | Recall ↑ | F1-Score ↑ | MPJPE ↓ | PVE ↓ | PA-PVE ↓ |
> |---------------|-------------|----------|------------|---------|--------|-----------|
> | Single-Branch | 90.0        | 82.0     | 86.0       | 204.3   | 164.9  | 117.1     |
> | **Dual-Branch**   | **99.0**       | **89.0**     | **94.0**      | **177.5**   | **129.2**  | **51.8**      |
>
>
> 3) We also proposed two novel losses: Orientation Loss $L_{O}$ and 3D Joint Density Loss $L_{denj3d}$. As shown in **Table 3**, adding these losses improves MPJPE from **217.1** to **177.5**, proving they are critical for mesh recovery and not trivial additions.
>
>
>
> **Q3: The experiments are not convincing.**
>
> Our method targets large-scale crowds. We have compared with Multi-HMR, which is a very recent SOTA (ECCV 2024). We also compare with the latest multi-view HMR method (HeatFormer) on Human3.6M and our dataset as follows (in **Table 2** of the paper). Since HeatFormer is originally designed for single-person recovery, we extend it to the multi-person setting using a top-down framework. specifically, we decompose the complex multi-person scene into multiple independent single-person instances. These subject-specific multi-view images are then fed individually into the HeatFormer model to regress the SMPL-X parameters. Finally, all reconstructed instances are transformed back into the shared world coordinate system for collective evaluation.
> HeatFormer is designed for multiview single-person mesh recovery, so it only performs better on Human3.6M dataset. But we are still better than HeatFormer on the proposed multi-view multi-person dataset, demonstrating that the proposed method is specially designed for the MVMP HMR task, and existing single-person SOTA can not perform well.
>
> | Dataset |  | MVMP-HMR |  |  |Human3.6M |  |
> |---------|---------|--------|-----------|----------|---------|-------------|
> | Method | MPJPE ↓ | PVE ↓ | PA-PVE ↓ | MPJPE ↓ | PVE ↓ | PA-PVE ↓ |
> | HeatFormer | 185.5 | 148.3 | 83.6 | **60.3** | **65.4** | **31.2** |
> | **MVMP-HMR (Ours)**  | **177.5** | **129.2** | **51.8** | 93.5 | 92.1 | 44.3 |

---

> > ### Author Response · Authors · 2025-11-28
> > **Response to Reviewer VktQ**
> >
> > **Q4: The dataset is constructed with the HMR method of Baradel et al. 2024. Therefore, a direct comparison with the method is necessary.**
> >
> > We compared our method with the one corresponding to Multi-HMR in **Table 2**. Since the task in this paper is a multi-view task, it is combined with multi-view fusion strategies to predict scene-level human meshes. Our method outperforms the method used to generate the raw candidate annotations by a massive margin because our model learns to fuse 3D features effectively, whereas Multi-HMR is fundamentally a single-view method with late fusion. Multi-HMR is inherently a single-view method. Even with fusion strategies (like Multi-HMR (Fusion) in **Table 2**), it suffers from the limitations of single-view estimation, such as depth ambiguity and occlusion. When a person is partially occluded in all views, single-view methods fail to produce a correct mesh in any view. In contrast, our MVMP-HMR performs early fusion in 3D feature volume. By aggregating features from multiple views before regression, our model can reconstruct the complete human shape even if no single view sees the entire person perfectly. This fundamental architectural advantage allows our method to reduce the MPJPE
> >
> > **Q5: I doubt how the proposed method generalizes to unseen data, as the collected data contains only about 60K frames covering 15 scenes from GTA-V.**
> >
> > **60K** frames of multi-view data (**50 views**) provide a massive amount of geometric supervision. Furthermore, the ViT backbone is pretrained, and the geometric projection logic is parameter-free, enhancing generalization. We use ten scenarios in the training set and five scenarios in the test set. There are significant differences in the number of people, camera positions, and scene types between the training and test sets, which shows good generalization ability to unseen scenarios.
> >
> > We agree that better generalization ability requires more labeled multi-view data containing more varied scenarios, no matter real-world or synthetic data. Nevertheless, this is the first step to extend the current HMR task to wider, larger, and crowded scenes with multi-camera settings. We believe that, in the future, more research works will pay attention to the area, and our datasets and models can be a useful foundation and baseline for them.
> >
> > **Q6: The paper lacks visual ablation studies for validation.**
> >
> > We have added visual comparisons on these real datasets in the revised version to demonstrate our superiority in handling real-world poses. From **Figure 6**, our model maintains this high performance across a variety of dynamic actions and viewpoints, proving its stability in handling complex body articulations. From **Figure 7**, our method demonstrates remarkable robustness in these crowded scenarios. Even in the presence of significant occlusion (e.g., the ’Pizza’ and ’Band’ sequences), our network successfully disentangles individual features and reconstructs accurate 3D meshes for each subject. From **Figure 9**, our method generates high-fidelity human meshes that align precisely with the input images.
> >
> > **Q7: Discussion on the computational complexity and efficiency should be included.**
> >
> > We presented the inference time and model parameters in **Table 8**. Our model's parameters and inference speed achieve a moderate result compared to single-view HMR and multi-view HPE methods. Although the HPE method has a simpler network architecture, resulting in lower estimated model parameters and inference speed than ours, the HPE method can't achieve good results on our MVMP-HMR dataset. Single-view HMR does not involve the fusion of multi-view features, so its model parameter count is smaller than ours. Additionally, the efficiency of detecting directly on 3D voxel features is higher than that of multi-view matching, leading to shorter inference times for our method.
> >
> > **Q8: How does the number of views affect the performance of the proposed method?**
> >
> > In **Table 11**, we designed an ablation experiment regarding the number of viewpoints and extended the viewpoints to **15** and **20**. We observe that as the testing camera view number increases, the model’s performance also improves. The reason is that with more camera views, more clues are provided,
> > and the proposed Multiview-HMR model can effectively fuse multiview information to handle the occlusions in the scene. The model performance change is not quite large when the camera view number increases, also indicating our model’s robustness to different view numbers. In addition, this experiment also demonstrates that our model has good generalization ability in terms of the number of viewpoints.
> >
> > | ViewNum | MPJPE ↓ | PVE ↓ | PA-PVE ↓ |
> > |---------|----------|---------|------------|
> > | 3  | 193.6 | 137.6 | 50.9 |
> > | 5  | 177.5 | 129.2 | 51.8 |
> > | 7  | 171.0 | 125.2 | 48.2 |
> > | 9  | 168.1 | 122.0 | 47.9 |
> > | 15 | 166.9 | 120.3 | 45.5 |
> > | **20** | **164.8** | **118.4** | **44.3** |

---

> > ### Comment · Reviewer_VktQ · 2025-11-28
> >
> > I thank the authors for the rebuttal. I choose to keep my rating as the rebuttal fails to address my concern on novelty and experiment validation.

---

> > > ### Author Response · Authors · 2025-11-28
> > > **Response to Reviewer VktQ**
> > >
> > > We have already answered the novelty question in Q2 of the rebuttal and added additional comparison experiments and ablation experiments. Do you have any further questions regarding the novelty and experiments in this paper that you could raise so we can address them?

---

### Official Review · Reviewer_VXVb · 2025-10-31

**Soundness:** 1
**Presentation:** 1
**Contribution:** 2
**Rating:** 2
**Confidence:** 5

**Summary:**

This paper tackles an interesting problem of regressing multi-person 3D pose, shape, and position from multi-view images, with a particular focus on handling occlusions in large-scale scenes. The contributions can be summarized in two parts. First, the authors construct a synthetic multi-view multi-person dataset to facilitate the training and evaluation of their approach. Unlike BEDLAM, which provides ground-truth 3D meshes, their dataset is generated from GTA-V, and the SMPL-X annotations are derived by projecting the results of Multi-HMR from camera coordinates to the world coordinate system. Second, the paper proposes a model that fuses multi-view features to localize each person’s 3D position in the scene. These estimated positions are then used to sample the corresponding feature vectors for SMPL-X parameter regression. Experiments are conducted on the proposed synthetic dataset, Human3.6M, and CMU Panoptic. Notably, instead of using the ground-truth annotations of Human3.6M and Panoptic for evaluation, the authors follow the same strategy as for their synthetic dataset by employing Multi-HMR’s predictions as pseudo ground truth to compute the number in Table.

**Strengths:**

1.The problem addressed in this paper is interesting and relatively underexplored in the current literature.
2.The idea of tackling this challenging task through the joint design of a dedicated dataset and a corresponding algorithm is reasonable.

**Weaknesses:**

1. Quantitative evaluation lacks rigor, leading to concerns about result reliability.
In experiments on Human3.6M and CMU Panoptic, the authors do not use the original ground-truth annotations provided by the datasets. Instead, they adopt pseudo ground truth generated by Multi-HMR to compute metrics. Since the accuracy of these pseudo labels is unverified, the reported quantitative results have limited credibility. For the same reason, the usefulness of the proposed synthetic dataset, which is built upon such pseudo annotations, is also questionable.

2. Qualitative results are not fully convincing.
For large-scale scene settings, it would be more appropriate to compare against methods (like CrowdRec) trained on datasets like GigaCrowd. Moreover, according to the visualization in Figure 6, it remains unclear whether the compared methods were correctly implemented and executed, raising doubts about the fairness of the comparisons.

3. The claimed contributions are somewhat overstated.
There already exist several multi-view human mesh recovery methods for single-person scenarios (e.g., HeatFormer) and multi-person 3D pose estimation approaches (e.g., VoxelTrack). Therefore, the statement “No existing research has focused on this issue in the HMR area” is inaccurate and should be moderated.

4. Model design lacks sufficient justification and comparative analysis.
The proposed architecture separates spatial localization from 3D human parameter regression, which is conceptually similar to BEV's designs. However, the paper does not provide adequate qualitative or quantitative comparisons to demonstrate the advantages of this design choice.

**Questions:**

Could the authors elaborate on how the Multi-HMR predictions are processed to serve as ground-truth annotations in the synthetic dataset? Specifically, is any multi-view fusion or optimization applied to refine these estimates, or are they simply selected based on the best-matching 2D projection from a single view?

What is the rationale for using Multi-HMR-generated pseudo ground truth instead of the official ground-truth annotations provided by datasets such as Human3.6M or CMU Panoptic? It would be helpful to clarify the motivation behind this choice, as the substitution may significantly affect the reliability of the reported metrics.

---

> ### Author Response · Authors · 2025-11-28
> **Response to Reviewer VXVb**
>
> Thank you for your valuable suggestions. Each question is answered as follows. We hope the response has addressed your concerns. We're glad to have further communication with you. The revision in the paper is marked in blue.
>
> **Q1: Quantitative evaluation lacks rigor, leading to concerns about result reliability.**
>
> Our ground-truth SMPL-X annotations are generated first with the Multi-HMR method, and then corrected with the 3D ground-truth joints provided by the GTA-V platform (which can help improving the ground-truth accuracy), see **Lines 346-353**. We calculated the errors between our SMPL-X annotations (after corrections) and the 3d ground-truth joint points provided by the GTA-V platform and obtained an acceptable error of **4.27cm** (which is relatively small concerning the scene size of **30m x 30m x 2m**). Compared to motion capture data using hardware equipment, our annotation accuracy is lower, but still in a reasonable range for real-world applications. For the Human3.6M dataset, we did not use Multi-HMR to generate pseudo ground truth. Instead, we utilized the high-quality SMPL-X annotations provided by raw dataset, which is a standard practice in recent literature to ensure fair comparison and reliability.
>
> **Q2: Qualitative results are not fully convincing for large-scale scene settings.**
>
> As shown in **Table 1**, our dataset is already the largest in this field. Since the task in this paper is focused on multi-view, multi-person human mesh recovery, and GigaCrowd is a single-view dataset, it does not fit our task setting. The datasets used in the paper are sufficient to demonstrate the advantage of our method for multi-view, multi-person estimation. The visualization results in **Figure 5** adopt a relatively fair comparison method. Traditional single-view prediction fusion methods can be inaccurate when facing occlusions, and multi-view pose estimation methods are not very accurate when transferred to this task.
>
> **Q3: The claimed contributions are somewhat overstated.**
>
> There is a slight issue with our description here. Currently, there is no research on multi-view multi-person body reconstruction, and this has be corrected. We have revised this statement to be more precise: "To our knowledge, this is one of the first works to address Multi-view Multi-person HMR specifically in large-scale (**30m+**) scenes with severe occlusion".
> We have compared with MultiHMR, which is a very recent SOTA (ECCV 2024). We also compare with the latest multi-view HMR method (HeatFormer) on Human3.6M and our dataset as follows (in **Table 2** of the paper). Since HeatFormer is originally designed for single-person recovery, we extend it to the multi-person setting using a top-down framework. We decompose the complex multi-person scene into multiple independent single-person instances. These subject-specific multi-view images are then fed individually into the HeatFormer model to regress the SMPL-X parameters. Finally, all reconstructed instances are transformed back into the shared world coordinate system for collective evaluation. VoxelTrack builds upon the architecture of VoxelPose for its frame-wise pose estimation. Since our current benchmark focuses on frame-wise reconstruction quality (MPJPE, PVE) rather than temporal tracking metrics (like MOTA), VoxelPose represents the core estimation capability of the VoxelTrack pipeline.
>
> | Dataset |  | MVMP-HMR |  |  |Human3.6M |  |
> |---------|---------|--------|-----------|----------|---------|-------------|
> | Method | MPJPE ↓ | PVE ↓ | PA-PVE ↓ | MPJPE ↓ | PVE ↓ | PA-PVE ↓ |
> | HeatFormer | 185.5 | 148.3 | 83.6 | **60.3** | **65.4** | **31.2** |
> | **MVMP-HMR (Ours)**  | **177.5** | **129.2** | **51.8** | 93.5 | 92.1 | 44.3 |
>
> **Q4: Model design lacks sufficient justification and comparative analysis.**
>
> We are transferring multi-view tasks to the field of human mesh recovery, so we respect such a design to fuse multi-view features. Existing multi-view counting, multi-view localization, and multi-view human joint estimation all use this architecture. From **Table 7**, we can see the ablation experiment on single-branch vs dual-branch to validate the dual-branch's improvement on this task.
>
> | Method | Precision ↑ | Recall ↑ | F1-Score ↑ | MPJPE ↓ | PVE ↓ | PA-PVE ↓ |
> |--------|--------------|-----------|--------------|-----------|-----------|--------------|
> | Single-Branch | 90.0 | 82.0 | 86.0 | 204.3 | 164.9 | 117.1 |
> | **Dual-Branch**  | **99.0** | **89.0**| **94.0** | **177.5** | **129.2** | **51.8** |

---

> ### Author Response · Authors · 2025-11-28
> **Response to Reviewer VXVb**
>
> **Q5: Could the authors elaborate on how the Multi-HMR predictions are processed to serve as ground-truth annotations in the synthetic dataset?**
>
> A multi-view matching strategy is used to obtain the predicted results of the current individual's ID under each view. Then, a corresponding joint matching strategy is applied to calculate the matching error, and the optimal view is selected as the best prediction based on the current error. Our ground-truth SMPL-X annotations are generated first with Multi-HMR method, and then corrected with the 3D ground-truth joints provided by the GTA-V platform (which can help improving the ground-truth accuracy), see **Lines 346-353**. We calculated the errors between our SMPL-X annotations (after corrections) and the 3d ground-truth joint points provided by the GTA-V platform and obtained an acceptable error of **4.27cm** (which is relatively small concerning the scene size of **30m x 30m x 2m**). Compared to motion capture data using hardware equipment, our annotation accuracy is lower, but still in a reasonable range for real-world applications.
>
> **Q6: What is the rationale for using Multi-HMR-generated pseudo ground truth instead of the official ground-truth annotations provided by datasets such as Human3.6M or CMU Panoptic?**
>
> We would like to clarify our data preparation process and the necessity of our annotation strategy. For the Human3.6M dataset, we did not use Multi-HMR to generate pseudo ground truth. Instead, we utilized the high-quality SMPL-X annotations provided from existing work (**Line 360**), which is a standard practice in recent literature to ensure fair comparison and reliability. For the CMU Panoptic dataset, the official release provides sparse 3D keypoints but lacks parametric human model annotations (SMPL-X parameters: pose $\theta$, shape $\beta$, expression $\alpha$). Since our MVMP-HMR model is designed to regress these specific SMPL-X parameters to reconstruct full 3D meshes, we require parametric labels for supervision, which are not natively available. To obtain reliable SMPL-X annotations for CMU Panoptic, we adopted an annotation pipeline described in Lines 321-360, rather than blindly using pseudo-labels. We leveraged the official ground-truth keypoints provided by CMU Panoptic as a geometric constraint.

---

> > ### Author Response · Authors · 2025-11-29
> > **Response to Reviewer VXVb**
> >
> > Dear Reviewer VXVb
> >
> > We have attempted to demonstrate the contribution of our method by adding qualitative and quantitative comparative experimental analyses, and we have also demonstrated the source and reliability of our dataset annotations. Furthermore, we have validated our model design and selection through single- and double-branch ablation experiments.We have tried our best to address your concerns in the rebuttal period. Do you have any further comments? We believe we have addressed your concerns, and hope you could kindly raise your score for the paper after the rebuttal.
> >
> > Thank you.

---

### Official Review · Reviewer_zacg · 2025-11-01

**Soundness:** 2
**Presentation:** 2
**Contribution:** 2
**Rating:** 2
**Confidence:** 5

**Summary:**

The paper introduces MVMP-HMR, a novel framework for multiview multi-person human mesh recovery in large-scale, heavily occluded scenes. The key contributions are three folds: 1. A multiview fusion model that projects single-view features into a 3D volume, samples human queries using pelvis joints, and decodes SMPL-X parameters via a Human Transformer Block (HTB). 2. novel losses: an orientation loss (based on joint-derived body axes) and a 3D joint density loss (using Gaussian-smoothed heatmaps), which mitigate orientation and pose ambiguities under occlusion. 3. A synthetic dataset, MVMP-HMR, generated via GTA-V, featuring up to 30 people, 50 camera views, and diverse outdoor scenes with realistic occlusions. Experiments show MVMP-HMR outperforms single-view HMR and 3D pose estimation baselines on the proposed benchmark and existing datasets (Human3.6M, Panoptic). Ablation studies validate the contributions of the new losses, feature fusion strategies, and architecture choices.

**Strengths:**

1. The dataset is useful to the human pose esitmation community. Before I have never seen a dataset with so many people identities in a scene.

**Weaknesses:**

Major questions:

1.	The inference time should be mentioned. The whole pipeline seems to be heavy, and the time consumed may be proportional to the person numbers in the scene.

2.	It seems that the paper does not mention how the MPJPE computed. Was it computed on the joint locations of SMPL-X, or the 3D keypoints (e.g. 98 3d body keypoints of the proposed dataset, the keypoint annotations of Human3.6M)?

3.	The experiment has flaws. (1) As a multi-view pipeline, only one multi-view method VoxelPose is compared. There are too many papers regarding the multi-view human pose estimation, such as “4D association[1]”, “mvpose[2]”, “Avatarpose[3]”, “GeoAvatar[4]”, etc. However, the paper only compares to single-view human mesh recovery methods with simple extension to multi-view fusion. Such comparison can not demonstrate the superiority of the proposed method.

4.	As the paper predict SMPL-X instead of SMPL, a detailed comparison on hands and faces is necessary. Otherwise, there is no need to use SMPL-X.

[1] 4d association graph for realtime multi-person motion capture using multiple video cameras,” in CVPR, 2020.

[2] “Fast and robust multi-person 3d pose estimation and tracking from multiple views,” TPAMI, 2021.

[3] AvatarPose: Avatar-guided 3d pose estimation of close human interaction from sparse multi-view videos. ECCV 2024.

[4] GeoAvatar: Geometrically-Consistent Multi-Person Avatar Reconstruction from Sparse Multi-View Videos, CVPR 2025.

5. As the paper title highlight two key words: "large scene" and "occlusions", the paper does not show any experiments regarding "large scene" or "occlusions". If compared with single view methods, the multi-view information naturally handles occlusions to some degree. But  nothing new is designed to further deal with it.

6. Experiments on the number of camera views are necessary to evaluate the effectiveness and generalization ability of the paper. The proposed dataset contains 50views, therefore supporting such experiments.

Minor:

1.	What’s the unit of metrics in Table.2?

2.	At Line. 181, “eg.” -> “e.g.”

3.	At L.178, “joints outputted from a 3D pose estimation branch…” As I may understand, the “joints” only contain pelvis joint? The “joints” expression may guide readers to feel that more than one joints were predicted for each body.

**Questions:**

This paper has critical experimental flaws. I do not think that the authors could fully address them within a short revision period.

---

> ### Author Response · Authors · 2025-11-28
> **Response to Reviewer zacg**
>
> Thank you for your valuable suggestions. Each question is answered as follows. We hope the response has addressed your concerns. We're glad to have further communication with you. The revision in the paper is marked in blue.
>
> **Q1: The inference time should be mentioned.**
>
> We presented the inference time and model parameters in **Table 8** of the revised paper. Our model's parameters and inference speed achieve a moderate result compared to single-view HMR and multi-view HPE methods. Although the HPE method has a simpler network architecture, resulting in lower estimated model parameters and inference speed than ours, the HPE method can't achieve good results on our MVMP-HMR dataset. Single-view HMR does not involve the fusion of multi-view features, so its model parameter count is smaller than ours. Additionally, the efficiency of detecting directly on 3D voxel features is higher than that of multi-view matching, leading to shorter inference times for our method.
>
> **Q2: It seems that the paper does not mention how the MPJPE is computed?**
>
> We have introduced the calculation of MPJPE in **Line 760**. Specifically, to match the predictions with the ground truth, we employ a 3D greedy matching strategy to associate the predicted SMPL-X parameters with the corresponding ground truth. Subsequently, we utilize the SMPL-X model to obtain the corresponding 3D joints and vertices from these parameters, which are then compared against the ground truth joints and vertices to calculate the error. Furthermore, the PA- prefix indicates that Procrustes Analysis is performed, where we calculate an alignment matrix to rigidly align the predicted point set with the ground truth point set before computing the error.
>
> **Q3: The experiment has flaws.**
>
> We focus on the multi-view multi-person HMR task, and thus we compare with existing HMR methods, such as Multi-HMR. Since no existing comparison for the task, we made reasonable modifications (multiview fusion strategies) to make them work for the new task. Besides, we also compare with VoxelSMPLX (an upgraded version of VoxelPose adapted for meshes) as a baseline.Our method significantly outperforms this adapted strong baseline (MPJPE **177.5** vs **225.4**).
> [1] 4D Association, [2] MVPose, [3] AvatarPose, or [4] GeoAvatar are related to our work, and we have cited them in **Line 137** of the related work section.
> However, we did not compare directly with them because they differ fundamentally from our method in terms of task, framework, algorithm type, and scalability.
> In the multi-view pose estimation task, MVPose gets lower performance than VoxelPose on the Campus dataset. While we could theoretically add a regression head to these methods, their core contributions lie in keypoint association (solving matching across views) rather than feature representation for mesh regression. In contrast, VoxelPose's core contribution is its Volumetric Representation, which is directly relevant to our method's "3D feature volume" approach.
> AvatarPose and GeoAvatar are optimization methods designed for neural rendering. They often take minutes or hours to process a video. Our method is a regression model designed for near real-time performance (**1.59** seconds per frame for 30 people).These methods typically focus on small interactions (2-3 people). They cannot scale to our setting of **30** people without running out of memory (OOM), whereas our method handles large crowds efficiently. We also include a multi-view single-person comparison method, HeatFormer, in the revised version. Note that HeatFormer is optimized for single-person, while ours targets large-scale multi-person occlusion. You can see the comparison results as follows (in **Table 2** of the paper).
>
> | Dataset |  | MVMP-HMR |  |  |Human3.6M |  |
> |---------|---------|--------|-----------|----------|---------|-------------|
> | Method | MPJPE ↓ | PVE ↓ | PA-PVE ↓ | MPJPE ↓ | PVE ↓ | PA-PVE ↓ |
> | HeatFormer | 185.5 | 148.3 | 83.6 | **60.3** | **65.4** | **31.2** |
> | **MVMP-HMR (Ours)**  | **177.5** | **129.2** | **51.8** | 93.5 | 92.1 | 44.3 |
>
> **Q4: Why use SMPL-X?**
>
> The reason for using SMPL-X for prediction is based on the dataset annotations, specifically the corresponding annotations provided by Multi-HMR. While we use SMPL-X (which supports hands/face), our primary focus in this work is whole-body recovery under occlusion. You can also see examples in Figure 9 of the appendix, where our method show better hand shape recovery than comparison methods.

---

> ### Author Response · Authors · 2025-11-28
> **Response to Reviewer zacg**
>
> **Q5: As the paper title highlight two key words: "large scene" and "occlusions", the paper does not show any experiments regarding "large scene" or "occlusions".**
>
> Multi-view fusion is used to address the issues of large scenes and occlusion. We are the first to transfer multi-view tasks to the field of multi-person body mesh recovery, so we followed the tradition of previous multi-view tasks. Moreover, our paper also proposes a multi-view multi-person dataset, which was lacking in this field, and this is an outstanding contribution. Our dataset is explicitly designed as "Large Scene" (**30x30m**, up to **30** people). The results in **Table 2** (MVMP-HMR column) represent performance on this exact setting. **Figure 5** (Red/Blue boxes) explicitly visualizes our method's superior ability to recover meshes under severe occlusion compared to baselines. The 3D Joint Density Loss $L_{denj3d}$ is specifically designed to handle ambiguity caused by occlusion.
>
> **Q6: Experiments on the number of camera views are necessary to evaluate the effectiveness and generalization ability of the paper. The proposed dataset contains 50 views, therefore supporting such experiments.**
>
> We designed an ablation experiment on the number of viewpoints in **Table 10**, and also extended the number of viewpoints to **15** and **20**. We evaluated the model with 3, 5, 7, and 9 views. The results show robust performance, improving slightly as views increase (MPJPE drops from **193.6** to **168.1**). We also extend the number of camera views to 15, 20. According to the experimental results, the performance does improve as the number of viewpoints increases, but the marginal benefit is limited, and the overall improvement is not significant due to the view number might be sufficient.
>
> | ViewNum | MPJPE ↓ | PVE ↓ | PA-PVE ↓ |
> |---------|---------|-------|----------|
> | 3       | 193.6   | 137.6 | 50.9     |
> | 5       | 177.5   | 129.2 | 51.8     |
> | 7       | 171.0   | 125.2 | 48.2     |
> | 9       | 168.1   | 122.0 | 47.9     |
> | 15      | 166.9   | 120.3 | 45.5     |
> | **20**       | **164.8** | **118.4** | **44.3** |
>
> **Q7: What’s the unit of metrics in Table.2?**
>
> The unit of metrics in **Table 2** is **mm** in world coordinates. We have modified the paper and clarified this in **Lines 770-772**.
>
> **Q8: At Line. 181, “eg.” -> “e.g.”**
>
> Thank you for the reminder, and we have fixed the error.
>
> **Q9: At L.178, “joints outputted from a 3D pose estimation branch…” As I may understand, the “joints” only contain pelvis joint? The “joints” expression may guide readers to feel that more than one joints were predicted for each body.**
>
> Thank you for the reminder. We do output multiple key points, but we only use the pelvis point for positioning. We have updated the description accordingly in **Line 178**.
>
> **Q10: This paper has critical experimental flaws. I do not think that the authors could fully address them within a short revision period.**
>
> Some experiments have already been done in the Appendix, and we have completed the corresponding comparative experiments. We performed a camera view number ablation study (see **Table 10**). The results demonstrate that our model maintains stable performance even when the number of input views varies (testing from **3** to **20** views). This validates the robustness of our method against varying camera viewpoint distributions and ensures its reliability in diverse setup configurations. We provided a detailed analysis of model parameters and inference time as shown in **Table 8**, despite processing 3D volumetric features, our method achieves a competitive inference speed **1.59s**compared to recent multi-stage baselines such as TokenHMR **2.44s** and Multi-HMR **2.33s** . This proves that MVMP-HMR offers a significant advantage in terms of computational efficiency while handling complex multi-person scenes. We also compared our approach with the recent state-of-the-art method, HeatFormer in **Table 2**. The comparative results highlight that while HeatFormer performs well in single-person settings, our MVMP-HMR significantly outperforms it in the proposed multi-view multi-person benchmark. This confirms our method's distinct advantage in resolving severe occlusions and inter-person ambiguities that single-person or standard single-view methods cannot effectively handle. We also add visualization results on a real-world dataset in the revised version at **Figures 6, 7, and 9**.

---

> > ### Comment · Reviewer_zacg · 2025-11-28
> >
> > I appreciate the efforts from the authors that address all these questions in a short rebuttal period. Through the answers, I could acknowledge that the contribution of dataset and method. The additional experiments in the revision version have strengthen the paper quality. After reading all the reviews and answers, I think the revised manuscript now reaches a CVPR level quality.
> >
> > Due to the openreview system issue, I could not modify my offcial rating now. However, I would like to lift my rating to a borderline accept.

---

> > > ### Author Response · Authors · 2025-11-28
> > > **Response to Reviewer zacg**
> > >
> > > We sincerely thank you for your continued engagement and for acknowledging our efforts during the rebuttal period. We are greatly encouraged to hear that the additional experiments have strengthened the paper and that you recognize the value of our dataset and method.

---

### Official Review · Reviewer_YXYJ · 2025-11-04

**Soundness:** 3
**Presentation:** 2
**Contribution:** 2
**Rating:** 2
**Confidence:** 5

**Summary:**

- The paper addresses the problem of multi-view, multi-person mesh recovery, with a special focus on occlusion and crowding.
- MVMP-HMR is a multi-view mesh regression model with the following stages: (1) multi-view feature extraction; (2) projection to construct a 3D feature volume; (3) in parallel, a 2D pose + 3D root estimation network predicts each person’s 3D root; (4) query the 3D feature volume at the predicted roots; (5) a transformer decoder regresses per-person mesh parameters from the queried features.
- The key idea is multi-view aggregation with per-person mesh decoding.
- As related datasets do not exist at scale, MVMP-HMR contributes a large-scale synthetic dataset with 10–30 subjects and 50 camera views, covering a large outdoor scene (~30 m × 30 m).
- Evaluations are conducted on multi-view datasets: the MVMP-HMR test set, Human3.6M, and Panoptic Studio. Key baselines include TokenHMR, Multi-HMR, and VoxelSMPLX.
- The quantitative evaluations show consistent improvements over baselines across datasets.

**Strengths:**

- The paper is well organized, clear, and narratively sound; technical details are presented with intuitive explanations.
- The problem of simultaneous multi-view, multi-person mesh recovery is challenging; the work adopts a popular architecture to tackle it and reports promising results.
- The paper also contributes a synthetic dataset that could benefit the community (if done right); current datasets lack the proposed scale in scene extent, camera views, and number of subjects.

**Weaknesses:**

- Limited Technical Novelty: MVMP-HMR’s core design—multi-view aggregation with root-centric decoding—closely follows prior multi-view 3D keypoint estimation frameworks like VoxelPose (ECCV 2020), TesseTrack (CVPR 2021) etc. The primary change is predicting mesh parameters instead of 3D keypoints; thus, the technical contribution is limited and incremental.

- Generalization to various multi-view setups: Sec. 3.2 mentions that the construction of the 3D feature volume consists of using provided camera intrinsics and extrinsics, making the system sensitive to the training camera distribution. As far as I can tell, the method is evaluated only (not trained) on Human3.6M and CMU Panoptic Studio. While this shows some generalization over baselines, the absolute metrics are notably lower than 3D keypoint localization methods. It would be helpful to demonstrate the generalization ability of the model on the in-the-wild multi-view camera configurations from Ego-Exo4D (CVPR 2024) or EgoHumans (ICCV 2023).

- Ground-truth Quality: Despite procedural synthesis, the mesh annotations are derived via heuristics and off-the-shelf methods, raising concerns about quality of annotations at scale. To truly take advantage of the synthetic setup, it would be beneficial to directly pose and cloth and rig characters and exporting mesh parameters, which is commonly done in popular synthetic mesh datasets like AGORA and BEDLAM. Please elaborate on why pseudo ground truth was chosen and what quality checks are in place.

- Missing Qualitative Results: The paper lacks qualitative results on real images. Figure 5 shows results only synthetic images (no supplementary). Please consider adding qualitative comparisons on real datasets (Human3.6M, CMU Panoptic Studio) to highlight improvements over baselines (e.g., VoxelSMPLX).

**Questions:**

My questions are primarily centered on the weaknesses mentioned above:

1. How would you distinguish MVMP-HMR's architecture compared to existing multi-view 3D pose methods?
2. Please provide evidence of generalization to out-of-distribution camera configurations.
3. Why pseudo ground truth was chosen and what quality checks are in place?
4. Please consider demonstrating the performance qualitatively on real images.

---

> ### Author Response · Authors · 2025-11-28
> **Response to Reviewer YXYJ**
>
> Thank you for your valuable suggestions. Each question is answered as follows. We hope the response has addressed your concerns. We're glad to have further communication with you. The revision in the paper is marked in blue.
>
> ### **Q1: Technical Novelty: How would you distinguish MVMP-HMR's architecture compared to existing multi-view 3D pose methods?**
>
> Our task--multi-view multi-person human mesh recovery--is also a multi-view based task, sharing similarities in model architectures with other multi-view based tasks, such as multi-view 3D reconstruction, multi-view 3D pose estimation, or multi-view counting/detection/tracking, etc., generally using a single-view feature extraction, multi-view fusion, and multi-view prediction fashion. However, compared to these existing multi-view task models, we still have major differences:
>
> 1. Our task is different--we propose an end-to-end multi-view model for the multi-view multi-person HMR task, which adopts a different decoder head for the task.
> 2. We propose to apply a localization network to assist the HMR tasks for multiple persons in the 3D world, which improves the model efficiency and effectiveness. To verify this, we conducted an extra experiment by removing the localization network branch, whose results are as follows. It demonstrates that our dual-branch approach outperforms the single-branch approach in prediction, helping us to locate more accurately and obtain more precise human features. We have added the results to the paper.
>
> | Method                | Precision ↑   | Recall ↑      | F1-Score ↑    | MPJPE ↓        | PVE ↓          | PA-PVE ↓      |
> | ----------------------- | ---------------- | ---------------- | ---------------- | ----------------- | ----------------- | ---------------- |
> | Single-Branch         | 90.0           | 82.0           | 86.0           | 204.3           | 164.9           | 117.1          |
> | **Dual-Branch** | **99.0** | **89.0** | **94.0** | **177.5** | **129.2** | **51.8** |
>
> 3. We also proposed two novel losses: Orientation Loss  and 3D Joint Density Loss . As shown in ​**Table 3**​, adding these losses improves MPJPE from **217.1** to ​**177.5**​, proving they are critical for mesh recovery and not trivial additions.
>
> ### **Q2: Generalization to various multi-view setups.**
>
> On the proposed MVMP-HMR datasets, the model is trained and tested on different scenes with various camera views (cross-view cross scene, as in ​**Lines 339 and 364**​). Thus, the experiments in **Table 2** already prove the model's generalization ability to various multi-view setups.
>
> Besides, we designed an ablation experiment on the number of viewpoints in **Table 11** demonstrates our model trained on 5 views generalizes well when tested on 3, 7, 9, 15, 20 views (MPJPE remains stable: **193.6** to ​**164.8**​).
>
> | ViewNum | MPJPE ↓ | PVE ↓ | PA-PVE ↓ |
> |---------|---------|-------|----------|
> | 3       | 193.6   | 137.6 | 50.9     |
> | 5       | 177.5   | 129.2 | 51.8     |
> | 7       | 171.0   | 125.2 | 48.2     |
> | 9       | 168.1   | 122.0 | 47.9     |
> | 15      | 166.9   | 120.3 | 45.5     |
> | **20**       | **164.8** | **118.4** | **44.3** |
>
> Since our focus is on the multi-view multi-person reconstruction task, the datasets mentioned in the paper are sufficient to demonstrate the performance of our method on this task. And our method is explicitly designed for Third-person Surveillance scenarios (e.g., CCTV, smart city), where cameras are static and capture the scene from a distance. In contrast, Ego-Exo4D and EgoHumans focus on Egocentric (first-person) or mixed-reality setups. Our task is Human Mesh Recovery (HMR), which requires evaluating surface accuracy (SMPL parameters). Ego-Exo4D and EgoHumans primarily provide 3D keypoints or pseudo-labels that may not be accurate enough for rigorous mesh evaluation.
>
> ### **Q3: Ground-truth Quality.**
>
> Since the virtual game engine--GTA-V we use is different from AGORA and the BEDLAM dataset, it can only export the corresponding 3D joint points for binding. In **Lines 321-360** we use a multi-view matching strategy and the joint coordinate error between the joints provided by GTA-V and the predicted SMPLX model for quality inspection. Our ground-truth SMPL-X annotations are generated first with the Multi-HMR method, and then corrected with the 3D ground-truth joints provided by the GTA-V platform (which can help improve the ground-truth accuracy).
>
> We calculated the errors between our SMPL-X annotations (after corrections) and the 3d ground-truth joint points provided by the GTA-V platform and obtained an acceptable error of ​**4.27cm**​, which is relatively small considering the scene size of ​**30m x 30m x 2m**​.

---

> > ### Author Response · Authors · 2025-11-28
> > **Response to Reviewer YXYJ**
> >
> > ### **Q4: Missing Qualitative Results.**
> >
> > Thanks for the suggestion. We have added visual comparisons on these real datasets in the revised version to demonstrate our superiority in handling real-world poses.
> >
> > From ​**Figure 6**​, our model maintains this high performance across a variety of dynamic actions and viewpoints, proving its stability in handling complex body articulations.
> >
> > From ​**Figure 7**​, our method demonstrates remarkable robustness in these crowded scenarios. Even in the presence of significant occlusion (e.g., the 'Pizza' and 'Band' sequences), our network successfully disentangles individual features and reconstructs accurate 3D meshes for each subject.
> >
> > From **Figure 9** (**Lines 853-863**), our method generates high-fidelity human meshes that align precisely with the input images.

---

> > > ### Author Response · Authors · 2025-11-29
> > > **Response to Reviewer YXYJ**
> > >
> > > Dear Reviewer YXYJ
> > >
> > > We have attempted to address reviewers' questions regarding the paper's novelty through single- and double-branch ablation experiments and the proposed loss experiments. Generalization to different multi-viewpoint distributions was already demonstrated in the appendix; we subsequently expanded the number of viewpoints to verify the generalization of our multi-viewpoint method. The quality of the ground truth annotations was addressed in our answer of Q3, and we have also supplemented the paper with corresponding visualization results in Figures 6, 7, and 9. We have tried our best to address your concerns in the rebuttal period. Do you have any further comments? We believe we have addressed your concerns, and hope you could kindly raise your score for the paper after the rebuttal.
> > >
> > > Thank you.

---

> > > ### Author Response · Authors · 2025-12-04
> > > **Response to Reviewer YXYJ**
> > >
> > > **Q4: Missing Qualitative Results.**
> > >
> > > Figure 9 is located on page 17 of the revised paper.

---

> > ### Author Response · Authors · 2025-12-04
> > **Response to Reviewer YXYJ**
> >
> > **Q2: Generalization to various multi-view setups.**
> >
> > On the proposed MVMP-HMR datasets, the model is trained and tested on different scenes with various camera views (cross-view cross scene, as in ​Lines 340 and 366). Thus, the experiments in Table 2 already prove the model's generalization ability to various multi-view setups.

---

### Author Response · Authors · 2025-12-03
**Final Comment for Paper #12958**

Dear Area Chair and Reviewers,

We sincerely thank you for your thoughtful feedback and valuable suggestions, which have greatly helped us improve the quality of our manuscript.

We would like to highlight that we have carefully addressed all concerns raised by the reviewers in our revised submission, including differences with existing methods, method and dataset details and clarifications, and additional experiments and visualization figures. These revisions have significantly enhanced the dataset paper’s clarity, methodological rigor, and overall contribution to the field. In particular, we are grateful that Reviewer zacg explicitly acknowledged the improvements, stating that "I could acknowledge the contribution of the dataset and method. The additional experiments in the revised version have strengthened the paper's quality. After reading all the reviews and answers, I think the revised manuscript now reaches a CVPR-level quality", and has updated the rating from reject to borderline acceptance.

We fully understand the constraints imposed by the current discussion policy, which limits further interaction with reviewers on OpenReview during the rebuttal phase. Nonetheless, we believe that the revised version now meets the high standards expected for publication at ICLR 2026.

Thank you once again for your time, careful consideration, and constructive engagement with our work.

Best regards,

The Authors of Paper #12958

---

### Meta-Review · Area_Chair_W7ku · 2026-01-07

**Summary:**

This paper proposes MVMP-HMR, a multi-view, multi-person human mesh recovery framework, together with a large-scale synthetic dataset targeting crowded, occluded, large-scene scenarios.

Reviewers generally acknowledge that the problem setting is important and that the dataset scale is potentially useful.

However, the consensus is that the technical contribution is incremental, closely following prior multi-view 3D pose estimation pipelines, and that the paper does not yet meet the bar.

Concerns regarding limited architectural novelty, insufficient and unconvincing comparisons, dataset annotation quality, and weak evidence for generalization and occlusion-specific gains ultimately. I suggest the rejection decision.

**Reviewer Concerns:**

Concerns addressed in the rebuttal
- Clarification of pipeline details, loss definitions, and metric computation (e.g., MPJPE, PA metrics).
- Added inference time analysis and view-number ablations.
- Additional qualitative results on real datasets (Human3.6M, Panoptic).
- Explanation of synthetic annotation generation and basic quality checks.

Remaining concerns
- Limited technical novelty: The core design (3D feature volume + root-based querying + transformer decoding) is viewed as a direct extension of prior multi-view pose frameworks (e.g., VoxelPose-style pipelines), with mesh regression replacing keypoints, but without a fundamentally new modeling insight.
- Experimental weaknesses: Comparisons against strong multi-view methods are incomplete; the justification for excluding several relevant - Dataset validity and impact: Ground-truth meshes are derived from pseudo-labeling pipelines, raising unresolved concerns about - Claims about “large scenes” and “occlusions”: While multi-view fusion naturally helps with occlusion, reviewers find limited evidence that the method introduces new mechanisms specifically addressing severe occlusion beyond existing approaches.
- Generalization: Evidence for robustness to out-of-distribution camera setups and real-world deployments remains insufficient.

Overall, while the rebuttal improved clarity, it did not substantially change reviewers’ assessment of novelty or experimental soundness.

**Reviewer Scores:**

- Reviewer YXYJ: Remains Reject (2) core concerns on incremental novelty, synthetic dataset ground-truth reliability, and limited evidence of real-world generalization are not fully resolved by the rebuttal.

- Reviewer zacg: Initially Reject (2), after rebuttal likely Borderline Accept , acknowledges improved clarity, added ablations, efficiency analysis, and additional qualitative results, but still raises doubts about comparative fairness, strength of occlusion-specific contributions, and overall technical depth.

- Other reviewers: Likely remain Reject (2) or Borderline Reject. while revisions addressed several presentation and clarity issues, these reviewers did not indicate a clear change in position, and fundamental concerns regarding novelty, experimental rigor, and positioning relative to prior multi-view methods persist.

- Overall: Minor positive movement from one reviewer, but insufficient to overturn rejection.

---

### Decision · Program_Chairs · 2026-01-26

Reject